# Representation Balancing MDPs
# for Off-Policy Policy Evaluation

**Yao Liu**
Stanford University
yaoliu@stanford.edu

**Omer Gottesman**
Harvard University
gottesman@fas.harvard.edu

**Aniruddh Raghu**
Cambridge University
aniruddhraghu@gmail.com

**Matthieu Komorowski**
Imperial College London
matthieu.komorowski@gmail.com

**Aldo Faisal**
Imperial College London
a.faisal@imperial.ac.uk

**Finale Doshi-Velez**
Harvard University
finale@seas.harvard.edu

**Emma Brunskill**
Stanford University
ebrun@cs.stanford.edu

## Abstract

We study the problem of off-policy policy evaluation (OPPE) in RL. In contrast to prior work, we consider how to estimate both the individual policy value and average policy value accurately. We draw inspiration from recent work in causal reasoning, and propose a new finite sample generalization error bound for value estimates from MDP models. Using this upper bound as an objective, we develop a learning algorithm of an MDP model with a balanced representation, and show that our approach can yield substantially lower MSE in common synthetic benchmarks and a HIV treatment simulation domain.

## 1 Introduction

In reinforcement learning, off-policy (batch) policy evaluation is the task of estimating the performance of some *evaluation* policy given data gathered under a different *behavior* policy. Off-policy policy evaluation (OPPE) is essential when deploying a new policy might be costly or risky, such as in consumer marketing, healthcare, and education. Technically off-policy evaluation relates to other fields that study counterfactual reasoning, including causal reasoning, statistics and economics.

Off-policy batch policy evaluation is challenging because the distribution of the data under the behavior policy will in general be different than the distribution under the desired evaluation policy. This difference in distributions comes from two sources. First, at a given state, the behavior policy may select a different action than the one preferred by the evaluation policy—for example, a clinician may chose to amputate a limb, whereas we may be interested in what might have happened if the clinician had not. We never see the counterfactual outcome. Second, the distribution of future states—not just the immediate outcomes—is also determined by the behavior policy. This challenge is unique to sequential decision processes and is not covered by most causal reasoning work: for example, the resulting series of a patient's health states observed after amputating a patient's limb is likely to be significantly different than if the limb was not amputated.

Approaches for OPPE must make a choice about whether and how to address this data distribution mismatch. Importance sampling (IS) based approaches [16, 23, 8, 5, 10, 22] are typically unbiased and strongly consistent, but despite recent progress tend to have high variance—especially if the evaluation policy is deterministic, as evaluating deterministic policies requires finding in the data

sequences where the actions exactly match the evaluation policy. However, in most real-world applications deterministic evaluation policies are more common—policies are typically to either amputate or not, rather than a policy that that flips a biased coin (to sample randomness) to decide whether to amputate. IS approaches also often rely on explicit knowledge of the behavior policy, which may not be feasible in situations such as medicine where the behaviors results from human actions. In contrast, some model based approaches ignore the data distribution mismatch, such as by fitting a maximum-likelihood model of the rewards and dynamics from the behavioral data, and then using that model to evaluate the desired evaluation policy. These methods may not converge to the true estimate of the evaluation policy's value, even in the limit of infinite data [15]. However, such model based approaches often achieve better empirical performance than the IS-based estimators [10].

In this work, we address the question of building model-based estimators for OPPE that both *do* have theoretical guarantees and yield better empirical performance that model-based approaches that ignore the data distribution mismatch. Typically we evaluate the quality of an OPPE estimate $\widehat{V}^{\pi_e}(s_0)$, where $s_0$ is an initial state, by evaluating its mean squared error (MSE). Most previous research (e.g. [10, 22]) evaluates their methods using MSE for the average policy value (APV): $[\mathbb{E}_{s_0}\widehat{V}^{\pi_e}(s_0) - \mathbb{E}_{s_0}V^{\pi_e}(s_0)]^2$, rather than the MSE for individual policy values (IPV): $\mathbb{E}_{s_0}[\widehat{V}^{\pi_e}(s_0) - V^{\pi_e}(s_0)]^2$. This difference is crucial for applications such as personalized healthcare since ultimately we may want to assess the performance of a policy for an specific individual (patient) state.

Instead, in this paper we develop an upper bound of the MSE for individual policy value estimates. Note that this bound is automatically an upper bound on the average treatment effect. Our work is inspired by recent advances[19, 11, 12] in estimating conditional averaged treatment effects (CATE), also known as heterogeneous treatment effects (HTE), in the contextual bandit setting with a single (typically binary) action choice. CATE research aims to obtain precise estimates in the difference in outcomes for giving the treatment vs control intervention for an individual (state).

Recent work [11, 19] on CATE[1] has obtained very promising results by learning a model to predict individual outcomes using a (model fitting) loss function that explicitly accounts for the data distribution shift between the treatment and control policies. We build on this work to introduce a new bound on the MSE for individual policy values, and a new loss function for fitting a model-based OPPE estimator. In contrast to most other OPPE theoretical analyses (e.g. [10, 5, 22]), we provide a finite sample generalization error instead of asymptotic consistency. In contrast to previous model value generalization bounds such as the Simulation Lemma [13], our bound accounts for the underlying data distribution shift if the data used to estimate the value of an evaluation policy were collected by following an alternate policy.

We use this to derive a loss function that we can use to fit a model for OPPE for deterministic evaluation policies. Conceptually, this process gives us a model that prioritizes fitting the trajectories in the batch data that match the evaluation policy. Our current estimation procedure works for deterministic evaluation policies which covers a wide range of scenarios in real-world applications that are particularly hard for previous methods. Like recently proposed IS-based estimators [22, 10, 7], and unlike the MLE model-based estimator that ignores the distribution shift [15], we prove that our model-based estimator is asymptotically consistent, as long as the true MDP model is realizable within our chosen model class; we use neural models to give our model class high expressivity.

We demonstrate that our resulting models can yield substantially lower mean squared error estimators than prior model-based and IS-based estimators on a classic benchmark RL task (even when the IS-based estimators are given access to the true behavior policy). We also demonstrate our approach can yield improved results on a HIV treatment simulator [6].

## 2 Related Work

Most prior work on OPPE in reinforcement learning falls into one of three approaches. The first, importance sampling (IS), reweights the trajectories to account for the data distribution shift. Under mild assumptions importance sampling estimators are guaranteed to be both unbiased and strongly consistent, and were first introduced to reinforcement learning OPPE by Precup et al. [16]. Despite

recent progress (e.g.[23, 8]) IS-only estimators still often yield very high variance estimates, particularly when the decision horizon is large, and/or when the evaluation policy is deterministic. IS estimators also typically result in extremely noisy estimates for policy values of individual states. A second common approach is to estimate a dynamics and reward model, which can substantially reduce variance, but can be biased and inconsistent (as noted by [15]). The third approach, doubly robust estimators, originates from the statistics community [17]. Recently proposed doubly robust estimators for OPPE from the machine and reinforcement learning communities [5, 10, 22] have sometimes yielded orders of magnitude tighter estimates. However, most prior work that leverages an approximate model has largely ignored the choice of how to select and fit the model parameters. Recently, Farajtabar et al. [7] introduced more robust doubly robust (MRDR), which involves fitting a Q function for the model-value function part of the doubly robust estimator based on fitting a weighted return to minimize the variance of doubly robust. In contrast, our work learns a dynamics and reward model using a novel loss function, to estimate a model that yields accurate individual policy value estimates. While our method can be combined in doubly robust estimators, we will also see in our experimental results that directly estimating the performance of the model estimator can yield substantially benefits over estimating a Q function for use in doubly robust.

OPPE in contextual bandits and RL also has strong similarities with the treatment effect estimation problem common in causal inference and statistics. Recently, different kinds of machine learning models such as Gaussian Processes [1], random forests [24], and GANs [25] have been used to estimate heterogeneous treatment effects (HTE), in non-sequential settings. Schulam and Saria [18] study using Gaussian process models for treatment effect estimation in continuous time settings. Their setting differs from MDPs by not having sequential states. Most theoretical analysis of treatment effects focuses on asymptotic consistency rather than generalization error.

Our work is inspired by recent research that learns complicated outcome models (reward models in RL) to estimate HTE using new loss functions to account for covariate shift [11, 19, 2, 12]. In contrast to this prior work we consider the sequential state-action setting. In particular, Shalit et al. [19] provided an algorithm with a more general model class, and a corresponding generalization bound. We extend this idea from the binary treatment setting to sequential and multiple action settings.

## 3 Preliminaries: Notation and Setting

We consider undiscounted finite horizon MDPs, with finite horizon $H < \infty$, bounded state space $\mathcal{S} \subset \mathbb{R}^d$, and finite action space $\mathcal{A}$. Let $p_0(s)$ be the initial state distribution, and $T(s'|s,a)$ be the transition probability. Given a state action pair, the expectation of reward $r$ is $\mathbb{E}[r|x,a] = \bar{r}(x,a)$. Given $n$ trajectories collected from a stochastic behavior policy $\mu$, our goal is to evaluate the policy value of $\pi(s)$. We assume the policy $\pi(s)$ is deterministic. We will learn a model of both reward and transition dynamics, $\widehat{M} = \langle \widehat{r}(s,a), \widehat{T}(s',s,a) \rangle$, based on a learned representation. The representation function $\phi : \mathcal{S} \mapsto \mathcal{Z}$ is a reversible and twice-differentiable function, where $\mathcal{Z}$ is the representation space. $\psi$ is the reverse representation such that $\psi(\phi(s)) = s$. The specific form of our MDP model is: $\widehat{M}_\phi = \langle \widehat{r}(s,a), \widehat{T}(s',s,a) \rangle = \langle h_r(\phi(s),a), h_T(\phi(s'),\phi(s),a) \rangle$, where $h_r$ and $h_T$ is some function over space $\mathcal{Z}$. We will use the notation $\widehat{M}$ instead of $\widehat{M}_\phi$ later for simplicity.

Let $\tau = (s_0, a_0, \ldots, s_H)$ be a trajectory of $H + 1$ states and actions, sampled from the joint distribution of MDP $M$ and a policy $\mu$. The joint distributions of $\tau$ are: $p_{M,\mu}(\tau) = p_0(s_0) \prod_{t=0}^{H-1} [T(s_{t+1}|s_t, a_t)\mu(a_t|s_t)]$. Given the joint distribution, we denote the associated marginal and conditional distributions as $p_{M,\mu}(s_0), p_{M,\mu}(s_0, a_0), p_{M,\mu}(s_0|a_0)$ etc. We also have the joint, marginal and conditional, distributions $p_{M,\mu}^\phi(\cdot)$ based on the representation space $\mathcal{Z}$. We focus on the undiscounted finite horizon case, using $V_{M,t}^\pi(s)$ to denote the $t$-step value function of policy $\pi$.

## 4 Generalization Error Bound for MDP based OPPE estimator

Our goal is to learn a MDP model $\widehat{M}$ that directly minimizes a good upper bound of the MSE for the individual evaluation policy $\pi$ values: $\mathbb{E}_{s_0}[V_{\widehat{M}}^\pi(s_0) - V_M^\pi(s_0)]^2$. This model can provide value function estimates of the policy $\pi$ and be used as part of doubly robust methods.

In the on-policy case, the Simulation Lemma ( [13] and repeated for completeness in Lemma 1) shows that MSE of a policy value estimate can be upper bounded by a function of the reward and transition prediction losses. Before we state this result, we first define some useful notation.

**Definition 1.** *The square error loss function of value function, reward, transition are:*

$$\bar{\ell}_V(s, \widehat{M}, H-t) = \left(V^\pi_{\widehat{M}, H-t}(s) - V^\pi_{M, H-t}(s)\right)^2 \quad \bar{\ell}_r(s_t, a_t, \widehat{M}) = (\widehat{r}(s_t, a_t) - \bar{r}(s_t, a_t))^2$$

$$\bar{\ell}_T(s_t, a_t, \widehat{M}) = \left(\int_{\mathcal{S}} \left(\widehat{T}(s'|s_t, a_t) - T(s'|s_t, a_t)\right) V^\pi_{\widehat{M}, H-t-1}(s') ds'\right)^2 \qquad (1)$$

Then the Simulation lemma ensures that

$$\mathbb{E}_{s_0}\left[V^\pi_{\widehat{M}}(s_0) - V^\pi_M(s_0)\right]^2 \leq 2H \sum_{t=0}^{H-1} \mathbb{E}_{s_t, a_t \sim p_{M, \pi}}\left[\bar{l}_r(s_t, a_t, \hat{M}) + \bar{l}_T(s_t, a_t, \hat{M})\right], \qquad (2)$$

The right hand side can be used to formulate an objective to fit a model for policy evaluation. In off-policy case our data is from a different policy $\mu$, and one can get unbiased estimation of the RHS of Equation 2 by importance sampling. However, this will provide an objective function with high variance, especially for a long horizon MDP or a deterministic evaluation policy due to the product of IS weights. An alternative is to learn an MDP model by directly optimizing the prediction loss over our observational data, ignoring the covariate shift. From the Simulation Lemma this minimizes an upper bound of MSE of behavior policy value, but the resulting model may not be a good one for estimating the evaluation policy value. In this paper we propose a new upper bound on the MSE of the individual evaluation policy values inspired by recent work in treatment effect estimation, and use this as a loss function for fitting models.

Before proceeding we first state our assumptions, which are common in most OPPE algorithms:

1. Support of behavior policy covers the evaluation policy: for any state $s$ and action $a$, $\mu(a|s) = 0$ only if $\pi(a|s) = 0$.
2. Strong ignorability: there are no hidden confounders that influence the choice of actions other than the current observed state.

Denote a *factual* sequence to be a trajectory that matches the evaluation policy, $a_0 = \pi(s_0), \ldots, a_{t-1} = \pi(s_{t-1})$ as $a_{0:t-1} = \pi$. Let a *counterfactual* action sequence $a_{0:t-1} \neq \pi$ be an action sequence with at least one action that does not match $\pi(s)$. $p_{M, \mu}(\cdot)$ is the distribution over trajectories under $M$ and policy $\mu$. We define the $H - t$ step value error with respect to the state distribution given the factual action sequence.

**Definition 2.** $H - t$ *step value error is:* $\epsilon_V(\widehat{M}, H - t) = \int_{\mathcal{S}} \bar{\ell}_V(s_t, H-t) p_{M, \mu}(s_t|a_{0:t-1} = \pi) ds_t$

We use the idea of bounding the distance between representations given factual and counterfactual action sequences to adjust the distribution mismatch. Here the distance between representation distributions is formalized by Integral Probability Metric (IPM).

**Definition 3.** *Let $p, q$ be two distributions and let $G$ be a family of real-valued functions defined over the same space. The integral probability metric is:* $IPM_G(p, q) = \sup_{g \in G} \left|\int g(x)(p(x) - q(x)) dx\right|$

Some important instances of IPM include the Wasserstein metric where $G$ is 1-Lipschitz continuous function class, and Maximum Mean Discrepancy where $G$ is norm-1 function class in RKHS.

Let $p^{\phi, F}_{M, \mu}(z_t) = p^\phi_{M, \mu}(z_t|a_{0:t} = \pi)$ and $p^{\phi, CF}_{M, \mu}(z_t) = p^\phi_{M, \mu}(z_t|a_t \neq \pi, a_{0:t-1} = \pi)$, where $F$ and $CF$ denote factual and counterfactual. We first give an upper bound of MSE in terms of an expected loss term and then develop a finite sample bound which can be used as a learning objective.

**Theorem 1.** *For any MDP $M$, approximate MDP model $\widehat{M}$, behavior policy $\mu$ and deterministic evaluation policy $\pi$, let $B_{\phi, t}$ and $G_t$ be a real number and function family that satisfy the condition in Lemma 4. Then:*

$$\mathbb{E}_{s_0}\left[V^\pi_{\widehat{M}}(s_0) - V^\pi_M(s_0)\right]^2 \leq 2H \sum_{t=0}^{H-1} \left[B_{\phi, t} IPM_{G_t}\left(p^{\phi, F}_{M, \mu}(z_t), p^{\phi, CF}_{M, \mu}(z_t)\right)\right.$$

$$\left. + \int_{\mathcal{S}} \frac{1}{p_{M, \mu}(a_{0:t} = \pi)} \left(\bar{\ell}_r(s_t, \pi(s_t), \widehat{M}) + \bar{\ell}_T(s_t, \pi(s_t), \widehat{M})\right) p_{M, \mu}(s_t, a_{0:t} = \pi) ds_t\right] \qquad (3)$$

**(Proof Sketch)** The key idea is to use Equation 20 in Lemma 1 to view each step as a contextual bandit problem, and bound $\epsilon_V(\widehat{M}, H)$ recursively. We decompose the value function error into a one step reward loss, a transition loss and a next step value loss, with respect to the on-policy distribution. We can treat this as a contextual bandit problem, and we build on the method in Shalit et al.'s work [19] about binary action bandits to bound the distribution mismatch by a representation distance penalty term; however, additional care is required due to the sequential setting since the next states are also influenced by the policy. By adjusting the distribution for the next step value loss, we reduce it into $\epsilon_V(\widehat{M}, H - t - 1)$, allowing us recursively repeat this process for H steps. □

This theorem bounds the MSE for the individual evaluation policy value by a loss on the distribution of the behavior policy, with the cost of an additional representation distribution metric. The first IPM term measures how different the state representations are conditional on factual and counterfactual action history. Intuitively, a balanced representation can generalize better from the observational data distribution to the data distribution under the evaluation policy, but we also need to consider the prediction ability of the representation on the observational data distribution. This bound quantitatively describes those two effects about MSE by the IPM term and the loss terms. The re-weighted expected loss terms over the *observational* data distribution is weighted by the marginal action probabilities ratio instead of the conditional action probability ratio, which is used in importance sampling. The marginal probabilities ratio has lower variance than the importance sampling weights (See Appendix C.3).

One natural approach might be to use the right hand side of Equation 3 as a loss, and try to directly optimize a representation and model that minimizes this upper bound on the mean squared error in the individual value estimates. Unfortunately, doing so can suffer from two important issues. (1) The subset of the data that matches the evaluation policy can be very sparse for large $t$, and though the above bound re-weights data, fitting a model to it can be challenging due to the limited data size. (2) Unfortunately this approach ignores all the other data present that do not match the evaluation policy. If we are also learning a representation of the domain in order to scale up to very large problems, we suspect that we may benefit from framing the problem as related to transfer or multitask learning.

Motivated by viewing off-policy policy evaluation as a transfer learning task, we can view the source task as the evaluating the behavior policy, for which we have on-policy data, and view the target task as evaluating the evaluation policy, for which we have the high-variance re-weighted data from importance sampling. This is similar to transfer learning where we only have a few, potentially noisy, data points for the target task. Thus we can take the idea of co-learning a source task and a target task at the same time as a sort of regularization given limited data. More precisely, we now bound the OPPE error by an upper bound of the sum of two terms:

$$\underbrace{\mathbb{E}_{s_0}\left[V^\pi_{\widehat{M}}(s_0) - V^\pi_M(s_0)\right]^2}_{\text{MSE}_\pi} + \underbrace{\mathbb{E}_{s_0}\left[V^\mu_{\widehat{M}}(s_0) - V^\mu_M(s_0)\right]^2}_{\text{MSE}_\mu}, \qquad (4)$$

where we bound the former part using Theorem 1. Thus our upper bound of this objective can address the issues with separately using $\text{MSE}_\pi$ and $\text{MSE}_\mu$ as objective: compared with IS estimation of $\text{MSE}_\pi$, the "marginal" action probability ratio has lower variance. The representation distribution distance term regularizes the representation layer such that the learned representation would not vary significantly between the state distribution under the evaluation policy and the state distribution under the behavior policy. That reduces the concern that using $\text{MSE}_\mu$ as an objective will force our model to evaluate the behavior policy, rather than the evaluation policy, more effectively.

Our work is also inspired by treatment effect estimation in the casual inference literature, where we estimate the difference between the treated and control groups. An analogue in RL would be estimating the difference between the target policy value and the behavior policy value, by minimizing the MSE of policy difference estimation. The objective above is an upper bound of the MSE of policy difference estimator: $\frac{1}{2}\mathbb{E}_{s_0}\left[\left(V^\pi_{\widehat{M}}(s_0) - V^\mu_{\widehat{M}}(s_0)\right) - \left(V^\pi_M(s_0) - V^\mu_M(s_0)\right)\right]^2 \leq \text{MSE}_\pi + \text{MSE}_\mu$

We now bound Equation 4 further by finite sample terms. For the finite sample generalization bound, we first introduce a minor variant of the loss functions, with respect to the sample set.

**Definition 4.** *Let $r_t$ and $s'_t$ be an observation of reward and next step given state action pair $s_t, a_t$. Define the loss functions as:*

$$\ell_r(s_t, a_t, r_t, \widehat{M}) = (\widehat{r}(s_t, a_t) - r_t)^2 \tag{5}$$

$$\ell_T(s_t, a_t, s'_t, \widehat{M}) = \left( \int_{\mathcal{S}} \widehat{T}(s'|s_t, a_t) V^\pi_{\widehat{M}, H-t-1}(s') ds' - V^\pi_{\widehat{M}, H-t-1}(s'_t) \right)^2 \tag{6}$$

**Definition 5.** *Define the empirical risk over the behavior distribution and weighted distribution as:*

$$\widehat{R}_\mu(\widehat{M}) = \frac{1}{n} \sum_{i=1}^{n} \sum_{t=0}^{H-1} \ell_r(s_t^{(i)}, a_t^{(i)}, r^{(i)}, \widehat{M}) + \ell_T(s_t^{(i)}, a_t^{(i)}, s'^{(i)}_t, \widehat{M}) \tag{7}$$

$$\widehat{R}_{\pi,u}(\widehat{M}) = \frac{1}{n} \sum_{i=1}^{n} \sum_{t=0}^{H-1} \frac{\mathbb{1}(a_{0:t}^{(i)} = \pi)}{\widehat{u}_{0:t}} \left[ \ell_r(s_t^{(i)}, a_t^{(i)}, r^{(i)}, \widehat{M}) + \ell_T(s_t^{(i)}, a_t^{(i)}, s'^{(i)}_t, \widehat{M}) \right], \tag{8}$$

*where $n$ is the dataset size, $s_t^{(i)}$ is the state of the $t^{th}$ step in the $i^{th}$ trajectory, and $\widehat{u}_{0:t} = \sum_{i=1}^{n} \frac{\mathbb{1}(a_{0:t}^{(i)}=\pi)}{n}$.*

**Theorem 2.** *Suppose $\mathcal{M}_\Phi$ is a model class of MDP models based on representation $\phi$. For $n$ trajectories sampled by $\mu$, let $\ell_t(s_t, a_t, \widehat{M}_\phi) = \ell_r(s_t, a_t, r_t, \widehat{M}) + \ell_T(s_t, a_t, s'_t, \widehat{M})$, and $d_t$ be the pseudo-dimension of function class $\{\ell_t(s_t, a_t, \widehat{M}_\phi), \widehat{M}_\phi \in \mathcal{M}_\Phi\}$. Suppose $\mathcal{H}$ is the reproducing kernel Hilbert space induced by $k$, and $\mathcal{F}$ is the unit ball in it. Assume there exists a constant $B_{\phi,t}$ such that $\frac{1}{B_{\phi,t}} \ell_t(\psi(z), \pi(\psi(z)), \widehat{M}_\phi) \in \mathcal{F}$. With probability $1 - 3\delta$, for any $\widehat{M} \in \mathcal{M}_\Phi$:*

$$\mathbb{E}_{s_0} \left[ V^\pi_{\widehat{M}}(s_0) - V^\pi_M(s_0) | \widehat{M} \right]^2 \leq MSE_\pi + MSE_\mu \leq 2H\widehat{R}_\mu(\widehat{M}) + 2H\widehat{R}_{\pi,u}(\widehat{M})$$

$$+ 2H \sum_{t=0}^{H-1} B_{\phi,t} \left( IPM_\mathcal{F} \left( \widehat{p}_{M,\mu}^{\phi,F}(z_t), \widehat{p}_{M,\mu}^{\phi,CF}(z_t) \right) + \min \left\{ \mathcal{D}_\delta^\mathcal{F} \left( \frac{1}{\sqrt{m_{t,1}}} + \frac{1}{\sqrt{m_{t,2}}} \right), 2\nu \right\} \right)$$

$$+ 2H \sum_{t=0}^{H-1} \frac{\mathcal{C}_{n,\delta,t}^\mathcal{M}}{n^{3/8}} \left( \mathbb{V}[\frac{\mathbb{1}(a_{0:t} = \pi)}{\widehat{u}_{0:t}}, \ell_t] + \mathbb{V}[1, \ell_t] + \ell_{t,\max} \mathbb{V}[\frac{\mathbb{1}(a_{0:t} = \pi)}{u_{0:t}}, 1] \right) \tag{9}$$

$m_{t,1}$ and $m_{t,2}$ are the number of samples used to estimate $\widehat{p}_{M,\mu}^{\phi,F}(z_t)$ and $\widehat{p}_{M,\mu}^{\phi,CF}(z_t)$ respectively. $\mathcal{D}_\delta^\mathcal{F}$ is a function of the kernel $k$. $\mathcal{C}_{n,\delta,t}^\mathcal{M}$ is a function of $d_t$. $\mathbb{V}[w, \ell_t] = \max\{\sqrt{\mathbb{E}_{p_{M,\mu}}[w^2\ell_t^2]}, \sqrt{\mathbb{E}_{\widehat{p}_{M,\mu}}[w^2\ell_t^2]}\}$. $\ell_{t,\max} = \max_{s_t, a_t} |\ell_t(s_t, a_t)|$.

The first term is the empirical loss over the observational data distribution. The second term is a re-weighted empirical loss, which is an empirical version of the first term in Theorem 1. As said previously, this re-weighting has less variance than importance sampling in practice, especially when the sample size is limited. Theorem 3 in Appendix C.3 shows that the variance of this ratio is also no greater than the variance of IS weights. Our bound is based on the empirical estimate of the marginal probability $u_{0:t}$ and we are not required to know the behavior policy. Our method's independence of the behavior policy is a significant advantage over IS methods which are very susceptible to errors its estimation, as we discuss in appendix A. In practice, this marginal probability $u_{0:t}$ is easier to estimate than $\mu$ when $\mu$ is unknown. The third term is an empirical estimate of IPM, which we described in Theorem 1. We use norm-1 RKHS functions and MMD distance in this theorem and our algorithm. There are similar but worse results for Wasserstein distance and total variation distance [20]. $\mathcal{D}_\delta^\mathcal{F}$ measures how complex $\mathcal{F}$ is. It is obtained from concentration measures about empirical IPM estimators [20]. The constant $\mathcal{C}_{n,\delta,t}^\mathcal{M}$ measures how complex the model class is and it is derived from traditional learning theory results [4].

We compare our bound with the upper bound of model error for OPPE in [9]. In the corrected version of corollary 2 in [9], the upper bound of absolute error has a linear dependency on $\sqrt{\bar{\rho}_{1:H}}$ where $\bar{\rho}_{1:H}$ is an upper bound of the importance ratio, which is usually a dominant term in long horizon cases. As we clarified in last paragraph, the re-weighting weights in our bound, which are marginal action probability ratios, enjoy a lower variance than IS weights (See Appendix C.3).

# 5 Algorithm for Representation Balancing MDPs

Based on our generalization bound above, we propose an algorithm to learn an MDP model for OPPE, minimizing the following objective function:

$$\mathcal{L}(\widehat{M}_\phi; \alpha_t) = \widehat{R}_\mu(\widehat{M}_\phi) + \widehat{R}_{\pi,u}(\widehat{M}_\phi) + \sum_{t=0}^{H-1} \alpha_t \text{IPM}_{\mathcal{F}}\left(\widehat{p}_{M,\mu}^{\phi,F}(z_t), \widehat{p}_{M,\mu}^{\phi,CF}(z_t)\right) + \frac{\mathfrak{R}(\widehat{M}_\phi)}{n^{3/8}} \quad (10)$$

This objective is based on Equation 9 in Theorem 2. We minimize the terms in this upper bound that are related to the model $\widehat{M}_\phi$. Note that since $B_{\phi,t}$ depends on the loss function, we cannot know $B_{\phi,t}$ in practice. We therefore use a tunable factor $\alpha$ in our algorithm. $\mathfrak{R}(\widehat{M}_\phi)$ here is some kind of bounded regularization term of model that one could choose, corresponding to the model class complexity term in Equation 9. This objective function matches our intuition about using lower-variance weights for the re-weighting component and using IPM of the representation to avoid fitting the behavior data distribution.

In this work, $\phi(s)$ and $\widehat{M}_\phi$ are parameterized by neural networks, due to their strong ability to learn representations. We use an estimator of IPM term from Sriperumbudur et al. [21]. All terms in the objective function are differentiable, allowing us to train them jointly by minimizing the objective by a gradient based optimization algorithm.

After we learn an MDP by minimizing the objective above, we use Monte-Carlo estimates or value iteration to get the value for any initial state $s_0$ as an estimator of policy value for that state. We show that if there exists an MDP and representation model in our model class that could achieve:

$$\min_{\widehat{M}_\phi}\left(R_\mu(\widehat{M}_\phi) + R_{\pi,u}(\widehat{M}_\phi) + \sum_{t=0}^{H-1} \alpha_t \text{IPM}_{\mathcal{F}}\left(p_{M,\mu}^{\phi,F}(z_t), p_{M,\mu}^{\phi,CF}(z_t)\right)\right) = 0,$$

then $\lim_{n\to\infty} \mathbb{E}_{s_0}[V_{\widehat{M}_{\phi^*}}^\pi(s_0) - V_M^\pi(s_0)]^2 \to 0$ and estimator $V_{\widehat{M}_{\phi^*}}^\pi(s_0)$ is a consistent estimator for any $s_0$. See Corollary 2 in Appendix for detail.

We can use our model in any OPPE estimators that leverage model-based estimators, such as doubly robust [10] and MAGIC [22], though our generalization MSE bound is just for the model value.

# 6 Experiments

## 6.1 Synthetic control domain: Cart Pole and Montain Car

We test our algorithm on two continuous-state benchmark domains. We use a greedy policy from a learned Q function as the evaluation policy, and an $\epsilon$-greedy policy with $\epsilon = 0.2$ as the behavior policy. We collect 1024 trajectories for OPPE. In Cart Pole domain the average length of trajectories is around 190 (long horizon variant), or around 23 (short horizon variant). In Mountain Car the average length of trajectories is around 150. The long horizon setting (H>100) is challenging for IS-based OPPE estimators due to the deterministic evaluation policy and long horizon, which will give the IS weights high variance. Deterministic dynamics and long horizons are common in real-world domains, and most off policy policy evaluation algorithms struggle in such scenarios.

We compare our method RepBM, with two baseline approximate models (AM and AM($\pi$)), doubly robust (DR), more robust doubly robust (MRDR), and importance sampling (IS). The baseline approximate model (AM) is an MDP model-based estimator trained by minimizing the empirical risk, using the same model class as RepBM. AM($\pi$) is an MDP model trained with the same objective as our method but without the $\text{MSE}_\mu$ term. DR is a doubly robust estimator using our model and DR(AM) is a doubly robust estimator using the baseline model. MRDR [7] is a recent method that trains a Q function as the model-based part in DR to minimize the resulting variance. We include their Q function estimator (MRDR Q), the doubly robust estimator that combines this Q function with IS (MRDR).

The reported results are square root of the average MSE over 100 runs. $\alpha$ is set to 0.01 for RepBM. We report mean and individual MSEs, corresponding to MSEs of average policy value and individual policy value, $[\mathbb{E}_{s_0}\widehat{V}(s_0) - \mathbb{E}_{s_0}V(s_0)]^2$ and $\mathbb{E}_{s_0}[\widehat{V}(s_0) - V(s_0)]^2$ respectively. IS and DR methods re-weight samples, so their estimates for single initial states are not applicable, especially in continuous state space. A comparison across more methods is included in the appendix.

Table 1: Root MSE for Cart Pole

| Long Horizon | RepBM | DR | AM | DR(AM) | AM($\pi$) | MRDR Q | MRDR | IS |
|---|---|---|---|---|---|---|---|---|
| Mean | **0.4121** | 1.359 | 0.7535 | 1.786 | 41.80 | 151.1 | 202 | 194.5 |
| Individual | **1.033** | - | 1.313 | - | 47.63 | 151.9 | - | - |
| Short Horizon | RepBM | DR | AM | DR(AM) | AM($\pi$) | MRDR Q | MRDR | IS |
| Mean | 0.07836 | **0.02081** | 0.1254 | 0.0235 | 0.1233 | 3.013 | 0.258 | 2.86 |
| Individual | **0.4811** | - | 0.5506 | - | 0.5974 | 3.823 | - | - |

Table 2: Root MSE for Mountain Car

| | RepBM | DR | AM | DR(AM) | AM($\pi$) | MRDR Q | MRDR | IS |
|---|---|---|---|---|---|---|---|---|
| Mean | **12.31** | 135.8 | 17.15 | 141.6 | 72.61 | 135.4 | 172.7 | 149.7 |
| Individual | **31.38** | - | 36.36 | - | 79.46 | 138.1 | - | - |

**Representation Balancing MDPs outperform baselines for long time horizons.** We observe that MRDR variants and IS methods have high MSE in the long horizon setting. The reason is that the IS weights for 200-step trajectories are extremely high-variance, and MRDR whose objective depends on the square of IS weights, also fails. Compared with the baseline model, we can see that our method is better than AM for both the pure model case and when used in doubly robust. We also observe that the IS part in doubly robust actually hurts the estimates, for both RepBM and AM.

**Representation Balancing MDPs outperform baselines in deterministic settings.** To observe the benefit of our method beyond long horizon cases, we also include results on Cart Pole with a shorter horizon, by using weaker evaluation and behavior policies. The average length of trajectories is about 23 in this setting. Here, we observe that RepBM is still better than other model-based estimators, and doubly robust that uses RepBM is still better than other doubly robust methods. Though MRDR produces substantially lower MSE than IS, which matches the report in Farajtabar et al. [7], it still has higher MSE than RepBM and AM, due to the high variance of its learning objective when the evaluation policy is deterministic.

**Representation Balancing MDPs produce accurate estimates even when the behavior policy is unknown.** For both horizon cases, we observe that RepBM learned with no knowledge of the behavior policy is better than methods such as MRDR and IS that use the true behavior policy.

## 6.2 HIV simulator

We demonstrate our method on an HIV treatment simulation domain. The simulator is described in Ernst et al. [6], and consists of 6 parameters describing the state of the patient and 4 possible actions. The HIV simulator has richer dynamics than the two simple control domains above. We learn an evaluation policy by fitted Q iteration and use the $\epsilon$-greedy policy of the optimal Q function as the behavior policy.

We collect 50 trajectories from the behavior policy and learn our model with the baseline approximate model (AM). We compare the root average MSE of our model with the baseline approximate MDP model, importance sampling (IS), per-step importance sampling (PSIS) and weighted per-step importance sampling (WPSIS). The root average MSEs reported are averaged over 80 runs. We observe that RepBM has the lowest root MSE on estimating the value of the evaluation policy.

Table 3: Relative Root MSE for HIV

| | RepBM | AM | IS | PSIS | WPSIS |
|---|---|---|---|---|---|
| Mean | **0.062** | 0.067 | 0.95 | 0.273 | 0.146 |

## 7 Discussion and Conclusion

One interesting issue for our method is the effect of the hyper-parameter $\alpha$ on the quality of estimator. In the appendix, we include the results of RepBM across different values of $\alpha$. We find that our method outperforms prior work for a large range of alphas, for both domains. In both domains

we observe that the effect of IPM adjustment (non-zero $\alpha$) is less than the effect of "marginal" IS re-weighting, which matches the results in Shalit et al.'s work in the binary action bandit case [19].

To conclude, in this work we give an MDP model learning method for the individual OPPE problem in RL, based on a new finite sample generalization bound of MSE for the model value estimator. We show our method results in substantially smaller MSE estimates compared to state-of-the-art baselines in common benchmark control tasks and on a more challenging HIV simulator.

### Acknowledgments

This work was supported in part by the Harvard Data Science Initiative, Siemens, and a NSF CAREER grant.

## Footnotes

[1]Shalit et al. [19] use the term individual treatment effect (ITE) to refer to a criterion which is actually defined as CATE in most causal inference literature. We discuss the confusion about the two terms in the appendix B.

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
