[Supplementary Material]

# A   IS with approximate behavior policy

In this section, we include some theoretical and empirical results about the effect of using an estimated behavior policy in importance sampling, when the true behavior policy is not accessible.

**Proposition 1.** *Assume the reward is in $[0, R_{\max}]$. For any estimator $\widehat{\mu}(a|s)$ of the true behavior policy $\mu(a|s)$, let $V_{IS}(\hat{\mu})$ be the IS estimator using this estimated $\hat{\mu}(a|s)$ and $V_{IS}(\mu)$ be the IS estimator with true behavior policy. Both of the IS estimators are computed using $n$ trajectories that are independent from the data used to estimate $\widehat{\mu}(a|s)$. If the relative error of $\widehat{\mu}(a|s)$ is bounded by $\delta$: $\|\frac{\widehat{\mu}(a|s)-\mu(a|s)}{\mu(a|s)}\|_\infty \leq \delta$, then for any given dataset:*

$$|V_{IS}(\hat{\mu}) - V_{IS}(\mu)| \leq \max\left\{\left(\frac{1}{1-\delta}\right)^H - 1, 1 - \left(\frac{1}{1+\delta}\right)^H\right\} V_{IS}(\mu)$$

*The bias of $V_{IS}(\hat{\mu})$ is bounded by:*

$$|\mathbb{E}V_{IS}(\hat{\mu}) - v| \leq \max\left\{\left(\frac{1}{1-\delta}\right)^H - 1, 1 - \left(\frac{1}{1+\delta}\right)^H\right\} v,$$

*where $v$ is the true evaluation policy value.*

*Proof.*

$$
\begin{aligned}
|V_{IS}(\hat{\mu}) - V_{IS}(\mu)| &= \left| \frac{1}{n}\sum_{i=1}^{n}\prod_{t=0}^{H-1}\frac{\pi(a_t^{(i)}|s_t^{(i)})}{\hat{\mu}(a_t^{(i)}|s_t^{(i)})}R_{0:H-1}^{(i)} - \frac{1}{n}\sum_{i=1}^{n}\prod_{t=0}^{H-1}\frac{\pi(a_t^{(i)}|s_t^{(i)})}{\mu(a_t^{(i)}|s_t^{(i)})}R_{0:H-1}^{(i)} \right| \\
&\leq \frac{1}{n}\sum_{i=1}^{n}\left| \prod_{t=0}^{H-1}\frac{\pi(a_t^{(i)}|s_t^{(i)})}{\hat{\mu}(a_t^{(i)}|s_t^{(i)})}R_{0:H-1}^{(i)} - \prod_{t=0}^{H-1}\frac{\pi(a_t^{(i)}|s_t^{(i)})}{\mu(a_t^{(i)}|s_t^{(i)})}R_{0:H-1}^{(i)} \right| \qquad (11) \\
&\leq \frac{1}{n}\sum_{i=1}^{n}\left| \prod_{t=0}^{H-1}\frac{\mu(a_t^{(i)}|s_t^{(i)})}{\hat{\mu}(a_t^{(i)}|s_t^{(i)})} - 1 \right| \left| \prod_{t=0}^{H-1}\frac{\pi(a_t^{(i)}|s_t^{(i)})}{\mu(a_t^{(i)}|s_t^{(i)})}R_{0:H-1}^{(i)} \right| \qquad (12)
\end{aligned}
$$

According to the condition, for any $a^{(i)}$ and $s^{(i)}$, $1 - \delta \leq \frac{\widehat{\mu}(a_t^{(i)}|s_t^{(i)})}{\mu(a_t^{(i)}|s_t^{(i)})} \leq 1 + \delta$. Then

$$\frac{1}{1+\delta} \leq \frac{\mu(a_t^{(i)}|s_t^{(i)})}{\hat{\mu}(a_t^{(i)}|s_t^{(i)})} \leq \frac{1}{1-\delta},$$

and:

$$\left(\frac{1}{1+\delta}\right)^H \leq \prod_{t=0}^{H-1}\frac{\mu(a_t^{(i)}|s_t^{(i)})}{\hat{\mu}(a_t^{(i)}|s_t^{(i)})} \leq \left(\frac{1}{1-\delta}\right)^H,$$

So:

$$\left| \prod_{t=0}^{H-1}\frac{\mu(a_t^{(i)}|s_t^{(i)})}{\hat{\mu}(a_t^{(i)}|s_t^{(i)})} - 1 \right| \leq \max\left\{\left(\frac{1}{1-\delta}\right)^H - 1, 1 - \left(\frac{1}{1+\delta}\right)^H\right\}$$

Plug this into Equation 12:

$$
\begin{aligned}
|V_{IS}(\hat{\mu}) - V_{IS}(\mu)| &\leq \max\left\{\left(\frac{1}{1-\delta}\right)^H - 1, 1 - \left(\frac{1}{1+\delta}\right)^H\right\} \frac{1}{n}\sum_{i=1}^{n}\prod_{t=0}^{H-1}\frac{\pi(a_t^{(i)}|s_t^{(i)})}{\mu(a_t^{(i)}|s_t^{(i)})}R_{0:H-1}^{(i)} \\
&= \max\left\{\left(\frac{1}{1-\delta}\right)^H - 1, 1 - \left(\frac{1}{1+\delta}\right)^H\right\} V_{IS}(\mu) \qquad (13)
\end{aligned}
$$

Similarly, for the bias:

$$|\mathbb{E}V_{IS}(\hat{\mu}) - v| = |\mathbb{E}V_{IS}(\hat{\mu}) - \mathbb{E}V_{IS}(\mu)| \tag{14}$$

$$= \left| \mathbb{E} \prod_{t=0}^{H-1} \frac{\pi(a_t^{(0)}|s_t^{(0)})}{\hat{\mu}(a_t^{(0)}|s_t^{(0)})} R_{0:H-1}^{(0)} - \mathbb{E} \prod_{t=0}^{H-1} \frac{\pi(a_t^{(0)}|s_t^{(0)})}{\mu(a_t^{(0)}|s_t^{(0)})} R_{0:H-1}^{(0)} \right| \tag{15}$$

$$\leq \left| \prod_{t=0}^{H-1} \frac{\mu(a_t^{(0)}|s_t^{(0)})}{\hat{\mu}(a_t^{(0)}|s_t^{(0)})} - 1 \right| \left| \mathbb{E} \left[ \prod_{t=0}^{H-1} \frac{\pi(a_t^{(0)}|s_t^{(0)})}{\mu(a_t^{(0)}|s_t^{(0)})} R_{0:H-1}^{(0)} \right] \right| \tag{16}$$

$$\leq \max \left\{ \left( \frac{1}{1-\delta} \right)^H - 1, 1 - \left( \frac{1}{1+\delta} \right)^H \right\} v \tag{17}$$

$\square$

We bound the error of IS estimates by the relative error of behavior policy estimates. Proposition 3 from Farajtabar et al. [7] gave an expression for the bias when using an empirical estimate of behavior policy in IS. The result in Farajtabar et al. [7] is similar to this proposition, but the authors did not explicitly bound the bias by the error of behavior policy. Note that this bound increases exponentially with the horizon $H$, which shows the accumulated error effect of behavior policy error.

By using the tree MDP example in Jiang and Li [10] we can show that the order of magnitude $O(\exp(H))$ is tight: there exists an MDP and a policy estimator $\hat{\mu}$ with $\|(\hat{\mu} - \mu)/\mu\|_\infty = \delta$ such that the bias of $V_{IS}(\hat{\mu})$ is $O(\exp(H))$. Define a binary discrete tree MDP [10] as following: At each node in a binary tree, we can take two actions $a = 0, a = 1$, leading to the two next nodes with observations $o = 0, o = 1$. The state of a node is defined by the whole path to the root: $o_0 a_0 o_1 a_i \ldots o_h$. That means each node in the tree will have a unique state. The depth of the tree, as well as the horizon of the trajectories, is $H$. Only the leftmost leaf node (by always taking $a = 0$) has non-zero reward $r = 1$. Denote this state as the target state. The evaluation policy always takes action $a = 0$ and the behavior policy $\mu$ is a uniform random policy. Let the estimated policy $\hat{\mu}$ differ from $\mu$ with $-\delta/2$ in all the state-action pairs on the path to target state. That means the action probability in $\hat{\mu}$ is $1/2 - \delta/2$ for all state-action pairs on the path to target state. The IS estimator with $\mu$ has expectation 1 since it is unbiased. It is easy to verify that the IS estimator using $\hat{\mu}$ has expectation $(1-\delta)^{-H}$. Thus the bias is $O(\exp(H))$.

This result represents the worst-case upper bound on the bias of IS when using an estimated behaviour policy; the fact it is exponential in the trajectory length illustrates the problem when using IS without knowing the true behaviour policy. To support this result with an empirical example illustrating the challenge of using IS with an unknown behaviour policy for a real data distribution, consider Figure 1 which represents the error in OPPE (found using Per-Decision WIS) as we vary the accuracy of the behaviour policy estimation. Two different behaviour policies are considered. The domain used in this example is a continuous 2D map ($s \in \mathbb{R}^2$) with a discrete action space, $\mathcal{A} = \{1, 2, 3, 4, 5\}$, with actions representing a movement of one unit in one of the four coordinate directions or staying in the current position. Gaussian noise of zero mean and specifiable variance is added onto state of the agent after each action, to provide environmental stochasticity. An agent starts in the top left corner of the domain and receives a positive reward within a given radius of the top right corner, and a negative reward within a given radius of the bottom left corner. The horizon is set to be 15. A k-Nearest Neighbours (kNN) model is used to estimate the behaviour policy distribution, given a set of training trajectories. The accuracy of the model is varied by changing the number of trajectories available and the number of neighbours used for behaviour policy estimation.

This plot shows how IS suffers from very poor estimates with even slight errors in the estimated behaviour policy – average absolute errors of as small as 0.06 can incur errors of up over 50% in OPPE. This provides additional motivation for our approach – we do not require the behaviour policy to be known for OPPE, avoiding the significant errors incurred by using incorrectly estimated behaviour policies.

Figure 1: Plots showing the mean and standard deviation of the fractional error in OPE, $\frac{\hat{V}-V}{V}$, as a function of the average absolute error in behaviour policy estimation, $\frac{1}{n}\sum_{i=1}^{n}|\mu(a^{(i)}|s^{(i)}) - \hat{\mu}(a^{(i)}|s^{(i)})|$, for two different behaviour policies. The quality of OPE with IS has a very significant dependence on the accuracy of the behaviour policy estimation.

# B Clarification of CATE/HTE and ITE

In the causal inference literature[14], for an single unit $i$ with covariate (state) $x_i$, we observe $Y_i(1)$ if we give the unit treatment and $Y_i(0)$ if not. The Individual Treatment Effect (ITE) is defined as:

$$D_i = Y_i(1) - Y_i(0), \tag{18}$$

for this particular set of observations $Y_i, x_i$. However $Y_i(1)$ and $Y_i(0)$ cannot be observed at the same time, which makes ITE unidentifiable without strong additional assumptions. Thus the conditional average treatment effect (CATE), also known as heterogeneous treatment effect (HTE) is defined as:

$$\tau(x) = \mathbb{E}[Y_i(1) - Y_i(0)|x] \tag{19}$$

which is a function of $x$ and is identifiable. Shalit et al. [19] defined ITE as $\tau(x)$, which is actually named as CATE or HTE in most causal reasoning literature. So we use the name CATE/HTE to refer to this quantity and it is inconsistent with Shalit et al. 's work. We clarify it here so that it does confuse the reader.

# C Proofs of Section 4

## C.1 Proofs of Theorem 1 and Corollary 1

Before we prove Lemma 4 and Theorem 1, we need some useful lemmas and assumptions. We restate a well-known variant of Simulation Lemma [13] in finite horizon case here:

**Lemma 1.** *(Simulation Lemma with finite horizon case) Define that $V_{\widehat{M},0}^{\pi}(s) = V_{M,0}^{\pi}(s) = 0$. For any approximate MDP model $\widehat{M}$, any policy $\pi$, and $t = 0, \ldots, H-1$:*

$$V_{\widehat{M},H-t}^{\pi}(s_t) - V_{M,H-t}^{\pi}(s_t) = \mathbb{E}_{a_t \sim \pi}\left[\widehat{r}(s_t, a_t) - \bar{r}(s_t, a_t) + \int_{\mathcal{S}}\left(\widehat{T}(s'|s_t, a_t) - T(s'|s_t, a_t)\right)\right.$$

$$\left. V_{\widehat{M},H-t-1}^{\pi}(s')ds' + \int_{\mathcal{S}}T(s'|s_t, a_t)\left(V_{\widehat{M},H-t-1}^{\pi}(s') - V_{M,H-t-1}^{\pi}(s')\right)ds'\right] \tag{20}$$

*Then:*

$$V_{\widehat{M},H}^{\pi}(s) - V_{M,H}^{\pi}(s) = \mathbb{E}_{\pi,M}\sum_{t=0}^{H-1}[\widehat{r}(s_t, a_t) - \bar{r}(s_t, a_t)$$

$$+ \int_{\mathcal{S}}\left(\widehat{T}(s'|s_t, a_t) - T(s'|s_t, a_t)\right)V_{\widehat{M},H-t-1}^{\pi}(s')ds'|s_0 = s] \tag{21}$$

**Lemma 2.** *Let $J_\psi(z)$ be the absolute of the determinant of the Jacobian of $\psi(z)$. Then for any $z_t = \phi(s_t)$ and any sequence of actions $a_{0:t} = a_0, \ldots, a_t$:*

$$p_{M,\mu}^\phi(z_t|a_{0:t}) = J_\psi(z_t)p_{M,\mu}(\psi(z_t)|a_{0:t})$$

*Proof.* By the change of variable formula in a probability density function, we have:

$$p_{M,\mu}^\phi(z_t|a_{0:t}) = \frac{p_{M,\mu}^\phi(z_t, a_{0:t})}{p_{M,\mu}^\phi(a_{0:t})} = \frac{p_{M,\mu}(\psi(z_t), a_{0:t})J_\psi(z_t)}{p_{M,\mu}(a_{0:t})} = J_\psi(z_t)p_{M,\mu}(\psi(z_t)|a_{0:t})$$

$\square$

**Lemma 3.** *Let $p$ and $q$ be two distributions over the state space with the form of $p_{M,\mu}(s_t|a_{0:t})$(the action sequence $a_{0:t}$ might be different for p and q), and $p^\phi$ and $q^\phi$ be the corresponding distributions over the representation space. For any real valued function over the state space $f$, if there exists a constant $B_\phi > 0$ and a function class $G$ such that: $\frac{1}{B_\phi}f(\psi(z)) \in G$ then we have that*

$$\int_{\mathcal{S}} f(s)p(s)ds - \int_{\mathcal{S}} f(s)q(s)ds \le B_\phi \mathbf{IPM}_G(p^\phi, q^\phi)$$

*Proof.*

$$
\begin{align}
\int_{\mathcal{S}} f(s)p(s)ds - \int_{\mathcal{S}} f(s)q(s)ds &= \int_{\mathcal{S}} f(s)(p(s) - q(s))ds \tag{22}\\
&= \int_{\mathcal{Z}} f(\psi(z))(p(\psi(z)) - q(\psi(z)))J_\psi(z)dz \tag{23}\\
&= \int_{\mathcal{Z}} f(\psi(z))(p^\phi(z) - q^\phi(z))dz \tag{24}\\
&= B_\phi \int_{\mathcal{Z}} \frac{1}{B_\phi}f(\psi(z))(p^\phi(z) - q^\phi(z))dz \tag{25}\\
&\le B_\phi \left|\int_{\mathcal{Z}} \frac{1}{B_\phi}f(\psi(z))(p^\phi(z) - q^\phi(z))dz\right| \tag{26}\\
&\le B_\phi \sup_{g \in G} \left|\int_{\mathcal{Z}} g(z)(p^\phi(z) - q^\phi(z))dz\right| \tag{27}\\
&= B_\phi \mathrm{IPM}_G(p^\phi, q^\phi) \tag{28}
\end{align}
$$

$\square$

The following lemma recursively bounds $\epsilon_V(\widehat{M}, H - t)$ by $\epsilon_V(\widehat{M}, H - t - 1)$, whose result allows us to bound $\mathrm{MSE}_\pi = \epsilon_V(\widehat{M}, H)$.

The main idea to prove this is using Equation 20 from simulation lemma to decompose the loss of value functions into a one step reward loss, a transition loss and a next step value loss, with respect to the on-policy distribution. We can treat this as a contextual bandit problem, with the right side of Equation 20 as the loss function. For the distribution mismatch term, we follow the method in Shalit et al.'s work [19] about binary action bandits to bound the distribution mismatch by a representation distance penalty term. By converting the next step value error in the right side of Equation 20 into $\epsilon_V(\widehat{M}, H - t - 1)$, we can repeat this process recursively to bound the value error for H steps.

**Lemma 4.** *For any MDP $M$, approximate MDP model $\widehat{M}$, behavior policy $\mu$ and deterministic evaluation policy $\pi$, let $B_{\phi,t}$ and $G_t$ be a scalar and a function family such that:*

$$
\frac{1}{B_{\phi,t}}\left[\bar{\ell}_r(\psi(z_t), \pi(\psi(z_t)), \widehat{M}) + \bar{\ell}_T(\psi(z), \pi(\psi(z_t)), \widehat{M})\right.
$$
$$
\left. + \frac{1}{2(H - t - 1)}\int_{\mathcal{Z}} h_T(z'|\psi(z_t), \pi(\psi(z_t)))\bar{\ell}_V(\psi(z'), \widehat{M}, H - t - 1)dz'\right] \in G_t \quad (29)
$$

*Then for any $t \leq H - 1$:*

$$\epsilon_V(\widehat{M}, H - t) \leq (H - t) \left[ 2 \int_{\mathcal{S}} \left[ \bar{\ell}_r(s_t, \pi(s_t), \widehat{M}) + \bar{\ell}_T(s_t, \pi(s_t), \widehat{M}) \right] p_{M,\mu}(s_t | a_{0:t} = \pi) ds_t \right.$$

$$\left. + \frac{\epsilon_V(\widehat{M}, H - t - 1)}{H - t - 1} + 2 B_{\phi,t} IPM_{G_t} \left( p_{M,\mu}^\phi(z_t | a_{0:t} = \pi), p_{M,\mu}^\phi(z_t | a_t \neq \pi, a_{0:t-1} = \pi) \right) \right]$$

*Proof.* Note the recursive form in Lemma 1. We could treat RL as dealing with a contextual bandit problem at each step. Here we view the right side of recursive result in simulation lemma (restated here)

$$V_{\widehat{M}, H-t}^\pi(s_t) - V_{M, H-t}^\pi(s_t) = [\widehat{r}(s_t, \pi(s_t)) - \bar{r}(s_t, \pi(s_t))$$

$$+ \int_{\mathcal{S}} \left( \widehat{T}(s'|s_t, \pi(s_t)) - T(s'|s_t, \pi(s_t)) \right) V_{\widehat{M}, H-t-1}^\pi(s') ds'$$

$$+ \int_{\mathcal{S}} T(s'|s_t, \pi(s_t)) \left( V_{\widehat{M}, H-t-1}^\pi(s_t) - V_{M, H-t-1}^\pi(s_t) \right) ds' \right] \quad (30)$$

as a kind of square loss for a one-step prediction problem, and we bound the whole loss by recursively bounding these one-step losses. The key here is to find the recursive form of this square loss.

Recall that the definition of $\bar{\ell}_r, \bar{\ell}_T$ We will apply Cauchy-Schwarz inequality to bound Equation 30. Note that if $X_n = X_{n-1} + a_n + b_n$. Then $X_n^2 = (\frac{X_{n-1}}{\sqrt{n-1}} \sqrt{n-1} + \sqrt{2} a_n \frac{1}{\sqrt{2}} + \sqrt{2} b_n \frac{1}{\sqrt{2}})^2 \leq (\frac{X_{n-1}^2}{n-1} + 2a_n^2 + 2b_n^2)n$. By applying this to Equation 30, we have that

$$\epsilon_V(\widehat{M}, H - t) = \int_{\mathcal{S}} \bar{\ell}_V(s_t, H - t) p_{M,\mu}(s_t | a_{0:t-1} = \pi) ds_t \quad (31)$$

$$\leq (H - t) \int_{\mathcal{S}} \left[ 2\bar{\ell}_r(s_t, \pi(s_t), \widehat{M}) + 2\bar{\ell}_T(s_t, \pi(s_t), \widehat{M}) \right. \quad (32)$$

$$\left. + \frac{1}{H - t - 1} \int_{\mathcal{S}} T(s'|s_t, \pi(s_t)) \bar{\ell}_V(s, \widehat{M}, H - t - 1) ds' \right] p_{M,\mu}(s_t | a_{0:t-1} = \pi) ds_t$$

Note that:

$$p_{M,\mu}(s_t | a_{0:t-1} = \pi) = p_{M,\mu}(s_t, a_t = \pi | a_{0:t-1} = \pi) + p_{M,\mu}(s_t, a_t \neq \pi | a_{0:t-1} = \pi) \quad (33)$$

$$= p_{M,\mu}(s_t | a_{0:t} = \pi) p(a_t = \pi | a_{0:t-1} = \pi) \quad (34)$$

$$+ p_{M,\mu}(s_t | a_t \neq \pi, a_{0:t-1} = \pi) p(a_t \neq \pi | a_{0:t-1} = \pi) \quad (35)$$

Let $c_t = p(a_t = \pi | a_{0:t-1} = \pi)$ then $1 - c_t = p(a_t \neq \pi | a_{0:t-1} = \pi)$. Then

$$\epsilon_V(\widehat{M}, H - t)$$

$$\leq c_t(H - t) \int_{\mathcal{S}} \left[ 2\bar{\ell}_r(s_t, \pi(s_t), \widehat{M}) + 2\bar{\ell}_T(s_t, \pi(s_t), \widehat{M}) \right.$$

$$\left. + \frac{1}{H - t - 1} \int_{\mathcal{S}} T(s'|s_t, \pi(s_t)) \bar{\ell}_V(s, \widehat{M}, H - t - 1) ds' \right] p_{M,\mu}(s_t | a_{0:t} = \pi) ds_t$$

$$+ (1 - c_t)(H - t) \int_{\mathcal{S}} \left[ 2\bar{\ell}_r(s_t, \pi(s_t), \widehat{M}) + 2\bar{\ell}_T(s_t, \pi(s_t), \widehat{M}) + \frac{1}{H - t - 1} \right.$$

$$\left. \int_{\mathcal{S}} T(s'|s_t, \pi(s_t)) \bar{\ell}_V(s, \widehat{M}, H - t - 1) ds' \right] p_{M,\mu}(s_t | a_t \neq \pi, a_{0:t-1} = \pi) ds_t \quad (36)$$

Let

$$f(s_t) = 2\bar{\ell}_r(s_t, \pi(s_t), \widehat{M}) + 2\bar{\ell}_T(s_t, \pi(s_t), \widehat{M}) + \frac{1}{H - t - 1} \int_{\mathcal{S}} T(s'|s_t, \pi(s_t)) \bar{\ell}_V(s, \widehat{M}, H-t-1) ds'$$

We are going use Lemma 3 to bound the difference between $\int_{\mathcal{S}} f(s_t) p_{M,\mu}(s_t | a_{0:t} = \pi) ds_t$ and $\int_{\mathcal{S}} f(s_t) p_{M,\mu}(s_t | a_t \neq \pi, a_{0:t-1} = \pi) ds_t$. Let $G_t$ and $B_{\phi,t} > 0$ be a function class and a constant

satisfying that:

$$\frac{1}{B_{\phi,t}} \left[ \bar{\ell}_r(\psi(z_t), \pi(\psi(z_t)), \widehat{M}) + \bar{\ell}_T(\psi(z), \pi(\psi(z_t)), \widehat{M}) \right.$$

$$\left. + \frac{1}{2(H-t-1)} \int_{\mathcal{Z}} h_T(z'|\psi(z_t), \pi(\psi(z_t)))\bar{\ell}_V(\psi(z'), \widehat{M}, H-t-1)dz' \right] \in G_t \quad (37)$$

Then following Lemma 3 we have that:

$$\int_S f(s_t)p_{M,\mu}(s_t|a_t \neq \pi, a_{0:t-1} = \pi)ds_t$$

$$\leq \int_S f(s_t)p_{M,\mu}(s_t|a_{0:t} = \pi)ds_t + 2B_{\phi,t}\text{IPM}_{G_t}\left( p_{M,\mu}^{\phi}(z_t|a_{0:t} = \pi), p_{M,\mu}^{\phi}(z_t|a_t \neq \pi, a_{0:t-1} = \pi) \right)$$

$$= \int_S f(s_t)p_{M,\mu}(s_t|a_{0:t} = \pi)ds_t + 2B_{\phi,t}\text{IPM}_{G_t}\left( p_{M,\mu}^{\phi,F}(z_t), p_{M,\mu}^{\phi,CF}(z_t) \right) \quad (38)$$

Substituting this into Equation 36 we have that

$$\epsilon_V(\widehat{M}, H-t)$$

$$\leq (H-t)\int_S f(s_t)p_{M,\mu}(s_t|a_{0:t} = \pi)ds_t + 2(1-c_t)(H-t)B_{\phi,t}\text{IPM}_{G_t}\left( p_{M,\mu}^{\phi,F}(z_t), p_{M,\mu}^{\phi,CF}(z_t) \right) \quad (39)$$

$$\leq (H-t)\int_{\mathcal{S}} \left[ 2\bar{\ell}_r(s_t, \pi(s_t), \widehat{M}) + 2\bar{\ell}_T(s_t, \pi(s_t), \widehat{M}) + \frac{\int_{\mathcal{S}} \widehat{T}(s'|s_t, \pi(s_t))\bar{\ell}_V(s, \widehat{M}, H-t-1)ds'}{H-t-1} \right]$$

$$p_{M,\mu}(s_t|a_{0:t} = \pi)ds_t + 2(1-c_t)(H-t)B_{\phi,t}\text{IPM}_{G_t}\left( p_{M,\mu}^{\phi,F}(z_t), p_{M,\mu}^{\phi,CF}(z_t) \right) \quad (40)$$

$$\leq 2(H-t)\int_{\mathcal{S}} \left[ \bar{\ell}_r(s_t, \pi(s_t), \widehat{M}) + \bar{\ell}_T(s_t, \pi(s_t), \widehat{M}) \right] p_{M,\mu}(s_t|a_{0:t} = \pi)ds_t$$

$$+ \frac{H-t}{H-t-1}\epsilon_V(\widehat{M}, H-t-1) + 2(H-t)B_{\phi,t}\text{IPM}_{G_t}\left( p_{M,\mu}^{\phi,F}(z_t), p_{M,\mu}^{\phi,CF}(z_t) \right) \quad (41)$$

Thus we finish the proof. $\qquad\square$

Iteratively applying this result for $t = 0, 1, \ldots, H$ we will have Theorem 1. Note that $\frac{1}{p_{M,\mu}(a_{0:t}=\pi)}p_{M,\mu}(s_t, a_{0:t} = \pi) = p_{M,\mu}(s_t|a_{0:t} = \pi)$.

**Theorem 1.** *(Restated) For any MDP $M$, approximate MDP model $\widehat{M}$, behavior policy $\mu$ and deterministic evaluation policy $\pi$, let $B_{\phi,t}$ and $G_t$ be a real number and function family that satisfies the condition in Lemma 4. Then:*

$$\mathbb{E}_{s_0}\left[ V_{\widehat{M}}^{\pi}(s_0) - V_M^{\pi}(s_0) \right]^2 \leq 2H \sum_{t=0}^{H-1} \left[ B_{\phi,t}\text{IPM}_{G_t}\left( p_{M,\mu}^{\phi,F}(z_t), p_{M,\mu}^{\phi,CF}(z_t) \right) \right.$$

$$\left. + \int_{\mathcal{S}} \frac{1}{p_{M,\mu}(a_{0:t} = \pi)}\left( \bar{\ell}_r(s_t, \pi(s_t), \widehat{M}) + \bar{\ell}_T(s_t, \pi(s_t), \widehat{M}) \right) p_{M,\mu}(s_t, a_{0:t} = \pi)ds_t \right]$$

For $\text{MSE}_\mu$, we can apply the simulation lemma to bound it by the reward and transition losses since the data distribution matches the policy $\mu$. Then we combine it with the theorem above. Note that $\text{MSE}_\pi \leq \text{MSE}_\pi + \text{MSE}_\mu$.

**Corollary 1.** *For any MDP $M$, approximate MDP model $\widehat{M}$, behavior policy $\mu$ and deterministic evaluation policy $\pi$, let $B_{\phi,t}$ and $G_t$ be a real number and function family that satisfies the condition*

*in Lemma 4. Let $u_{0:t} = p_{\mu,M}(a_{0:t} = \pi)$. Then:*

$$MSE_\pi \leq MSE_\pi + MSE_\mu$$

$$\leq 2H \sum_{t=0}^{H-1} \left[ \frac{1}{u_{0:t}} \int_{\mathcal{S}} \left( \bar{\ell}_r(s_t, \pi(s_t), \widehat{M}) + \bar{\ell}_T(s_t, \pi(s_t), \widehat{M}) \right) p_{M,\mu}(s_t, a_{0:t} = \pi) ds_t + \right.$$

$$\int_{\mathcal{S}} \sum_{a_t \in \mathcal{A}} \left( \bar{\ell}_r(s_t, a_t, \widehat{M}) + \bar{\ell}_T(s_t, a_t, \widehat{M}) \right) p_{\mu,M}(s_t, a_t) ds_t$$

$$\left. + B_{\phi,t} \mathit{IPM}_{G_t} \left( p_{M,\mu}^{\phi,F}(z_t), p_{M,\mu}^{\phi,CF}(z_t) \right) \right]$$

*Proof.* According to Lemma 1, the mean square error of estimating the behavior policy value could be written as:

$$\left[ \mathbb{E}_{s_0} V_{\widehat{M}}^\mu(s_0) - \mathbb{E}_{s_0} V_M^\mu(s_0) \right]^2$$

$$= \left[ \mathbb{E}_{\mu,M} \left( \sum_{t=0}^{H-1} \widehat{r}(s_t, a_t) - \bar{r}(s_t, a_t) + \int_{\mathcal{S}} \left( \widehat{T}(s'|s_t, a_t) - T(s'|s_t, a_t) \right) V_{\widehat{M}, H-t-1}^\pi(s') ds' \mid s_0 \right) \right]^2$$

$$\leq \mathbb{E}_{\mu,M} \left[ \left( \sum_{t=0}^{H-1} \widehat{r}(s_t, a_t) - \bar{r}(s_t, a_t) + \int_{\mathcal{S}} \left( \widehat{T}(s'|s_t, a_t) - T(s'|s_t, a_t) \right) V_{\widehat{M}, H-t-1}^\pi(s') ds' \right)^2 \right] \quad (42)$$

$$\leq \mathbb{E}_{\mu,M} \left[ 2H \sum_{t=0}^{H-1} \bar{\ell}_r(s_t, a_t, \widehat{M}) + \bar{\ell}_T(s_t, a_t, \widehat{M}) \right] \quad (43)$$

$$= 2H \sum_{t=0}^{H-1} \int_{\mathcal{S}} \sum_{a_t} \left( \bar{\ell}_r(s_t, a_t, \widehat{M}) + \bar{\ell}_T(s_t, a_t, \widehat{M}) \right) p_{\mu,M}(s_t, a_t) ds_t \quad (44)$$

The first step follows from Lemma 1. The second step follows from Jensen's inequality, and the third step follows from Cauchy-Schwarz inequality. By combining the results above with Theorem 1, we have that:

$$\mathbb{E}_{s_0} \left[ V_{\widehat{M}}^\pi(s_0) - V_M^\pi(s_0) \right]^2$$

$$\leq \mathbb{E}_{s_0} \left[ V_{\widehat{M}}^\pi(s_0) - V_M^\pi(s_0) \right]^2 + \mathbb{E}_{s_0} \left[ V_{\widehat{M}}^\mu(s_0) - V_M^\mu(s_0) \right]^2 \quad (45)$$

$$\leq 2H \sum_{t=0}^{H-1} \left[ \int_{\mathcal{S}} \left( \bar{\ell}_r(s_t, \pi(s_t), \widehat{M}) + \bar{\ell}_T(s_t, \pi(s_t), \widehat{M}) \right) p_{M,\mu}(s_t|a_{0:t} = \pi) ds_t \right.$$

$$+ \int_{\mathcal{S}} \sum_{a_t \in \mathcal{A}} \left( \bar{\ell}_r(s_t, a_t, \widehat{M}) + \bar{\ell}_T(s_t, a_t, \widehat{M}) \right) p_{\mu,M}(s_t, a_t) ds_t$$

$$\left. + B_{\phi,t} \mathrm{IPM}_{G_t} \left( p_{M,\mu}^{\phi,F}(z_t), p_{M,\mu}^{\phi,CF}(z_t) \right) \right] \quad (46)$$

$$= 2H \sum_{t=0}^{H-1} \left[ \frac{1}{u_{0:t}} \int_{\mathcal{S}} \left( \bar{\ell}_r(s_t, \pi(s_t), \widehat{M}) + \bar{\ell}_T(s_t, \pi(s_t), \widehat{M}) \right) p_{M,\mu}(s_t, a_{0:t} = \pi) ds_t \right.$$

$$\left. + \int_{\mathcal{S}} \sum_{a_t \in \mathcal{A}} \left( \bar{\ell}_r(s_t, a_t, \widehat{M}) + \bar{\ell}_T(s_t, a_t, \widehat{M}) \right) p_{\mu,M}(s_t, a_t) ds_t \right]$$

$$+ 2H \sum_{t=0}^{H-1} B_{\phi,t} \mathrm{IPM}_{G_t} \left( p_{M,\mu}^{\phi,F}(z_t), p_{M,\mu}^{\phi,CF}(z_t) \right) \quad (47)$$

$$\square$$

## C.2 Proof of Theorem 2

We showed in Theorem 1 that we can bound MSE by expected losses under the behavior policy distribution and an IPM term. In this section, we are going to further bound this by empirical losses and a generalization gap. We will firstly define some loss terms that are based on observations, instead of losses that are on expected values, $\bar{r}$ and $T(\cdot|s, a)$. Then we will introduce some lemmas that allow us to bound the generalization gap of weighted losses and IPM terms from previous works. Finally we will prove the finite sample MSE bound by putting these generalization gaps together.

**Definition 4.** *(Restated) Let $r_t$ and $s'_t$ be an observation of reward and next step given state action pair $s_t, a_t$. Define the loss function as:*

$$\ell_r(s_t, a_t, r_t, \widehat{M}) = (\widehat{r}(s_t, a_t) - r_t)^2$$

$$\ell_T(s_t, a_t, s'_t, \widehat{M}) = \left( \int_{\mathcal{S}} \left( \widehat{T}(s'|s_t, a_t) - \delta(s' - s'_t) \right) V^\pi_{\widehat{M}, H-t-1}(s')ds' \right)^2$$

$$= \left( \int_{\mathcal{S}} \widehat{T}(s'|s_t, a_t) V^\pi_{\widehat{M}, H-t-1}(s')ds' - V^\pi_{\widehat{M}, H-t-1}(s'_t) \right)^2$$

*where $\delta$ is the Dirac delta function.*

Actually the difference between $\ell$ and $\bar{\ell}$ can be captured by the variance of the reward and transition dynamics, which only depend on the underlying dynamics. The following definition and lemmas show that.

**Definition 6.** *Define the variance of t-th step reward and transition with respect to the state-action distribution $q(s_t, a_t)$ as:*

$$\sigma_{q,t} = \sigma_q(r) + \sigma_q(T) \qquad \sigma_{q,t}(r) = \int_{\mathcal{S}} \sum_{a_t} \int_{\mathcal{R}} (r - \bar{r}(s_t, a_t))^2 p_M(r|s_t, a_t) q(s_t, a_t) dr ds_t$$

$$\sigma_{q,t}(T) = \int_{\mathcal{S}} \sum_{a_t} \int_{\mathcal{S}} \left( \int_{\mathcal{S}} T(s'|s_t, a_t) V^\pi_{\widehat{M}, H-t-1}(s')ds' - V^\pi_{\widehat{M}, H-t-1}(s'_t) \right)^2 p_M(s'_t|s_t, a_t) q(s_t, a_t) ds'_t ds_t$$

**Lemma 5.** *(Variance decomposition)*

$$\int_{\mathcal{S}} \sum_{a_t} \left( \bar{\ell}_r(s_t, a_t, \widehat{M}) + \bar{\ell}_T(s_t, a_t, \widehat{M}) \right) q(s_t, a_t) ds_t = \int_{\mathcal{S}} \sum_{a_t} \left( \int_{\mathcal{R}} \ell_r(s_t, a_t, r, \widehat{M}) p(r|s_t, a_t) dr \right.$$

$$+ \left. \int_{\mathcal{S}} \ell_T(s_t, a_t, s'_t, \widehat{M}) p(s'_t|s_t, a_t, s'_t) ds'_t \right) q(s_t, a_t) ds_t - \sigma_{q,t}$$

*Proof.* Let's start with the $\ell_r$ and $\ell_T$ terms:

$$\ell_r(s_t, a_t, r_t, \widehat{M}) = (\widehat{r}(s_t, a_t) - r_t)^2 = (\widehat{r}(s_t, a_t) - \bar{r}(s_t, a_t))^2 + (\bar{r}(s_t, a_t) - r_t)^2$$
$$+ 2 \left( \widehat{r}(s_t, a_t) - \bar{r}(s_t, a_t) \right) \left( \bar{r}(s_t, a_t) - r_t \right) \quad (48)$$

Note that $\mathbb{E}\left[ \bar{r}(s_t, a_t) - r_t \right] = 0$ so the last term will be zero after we apply the integral. Then:

$$\int_{\mathcal{S}} \sum_{a_t} \int_{\mathcal{R}} \ell_r(s_t, a_t, r, \widehat{M}) p(r|s_t, a_t) q(s_t, a_t) dr ds_t = \int_{\mathcal{S}} \sum_{a_t} \bar{\ell}_r(s_t, a_t, \widehat{M}) q(s_t, a_t) ds_t + \sigma_{q,t}(r)$$

Similarly, for $\ell_T$ we have that:

$$\ell_T(s_t, a_t, s'_t, \widehat{M}) \tag{49}$$

$$= \left( \int_{\mathcal{S}} \widehat{T}(s'|s_t, a_t) V^\pi_{\widehat{M}, H-t-1}(s')ds' - V^\pi_{\widehat{M}, H-t-1}(s'_t) \right)^2 \tag{50}$$

$$= \left( \mathbb{E}_{s' \sim \widehat{T}} V^\pi_{\widehat{M}, H-t-1}(s') - V^\pi_{\widehat{M}, H-t-1}(s'_t) \right)^2 \tag{51}$$

$$= \left( \mathbb{E}_{\widehat{T}} V^\pi_{\widehat{M}, H-t-1}(s') - \mathbb{E}_T V^\pi_{\widehat{M}, H-t-1}(s') \right)^2 + \left( \mathbb{E}_T V^\pi_{\widehat{M}, H-t-1}(s') - V^\pi_{\widehat{M}, H-t-1}(s'_t) \right)^2$$

$$+ 2 \left( \mathbb{E}_{\widehat{T}} V^\pi_{\widehat{M}, H-t-1}(s') - \mathbb{E}_T V^\pi_{\widehat{M}, H-t-1}(s') \right) \left( \mathbb{E}_T V^\pi_{\widehat{M}, H-t-1}(s') - V^\pi_{\widehat{M}, H-t-1}(s'_t) \right) \tag{52}$$

Note that

$$\mathbb{E}_{s_t'\sim T}\left[\mathbb{E}_{s'\sim T}V_{\widehat{M},H-t-1}^{\pi}(s') - V_{\widehat{M},H-t-1}^{\pi}(s_t')\right] = \mathbb{E}_{s'\sim T}V_{\widehat{M},H-t-1}^{\pi}(s') - \mathbb{E}_{s_t'\sim T}V_{\widehat{M},H-t-1}^{\pi}(s_t') = 0$$

So the last term here will also be zero when we apply integral over $s_t'$ to $\ell_T(s_t, a_t, s_t', \widehat{M})$ and we have that:

$$\int_{\mathcal{S}}\sum_{a_t}\int_{\mathcal{S}}\ell_T(s_t, a_t, s_t', \widehat{M})p(s_t'|s_t, a_t, s_t')q(s_t, a_t)ds_t'ds_t$$

$$= \int_{\mathcal{S}}\sum_{a_t}\bar{\ell}_T(s_t, a_t, \widehat{M})q(s_t, a_t)ds_t + \sigma_{q,t}(T) \quad (53)$$

Thus we finished the proof by combining the $\ell_r$ part with $\ell_T$ part. $\qquad\square$

Now we are going to bound the expected value of $\ell_r$ and $\ell_T$ terms by the empirical mean of it. We restate our definition about empirical risk and add the definition about corresponding generalization risk:

**Definition 5.** *(Restate)*

$$R_\mu(\widehat{M}) = \sum_{t=0}^{H-1}\int_{\mathcal{S}}\sum_{a_t}\left(\int_{\mathcal{R}}\ell_r(s_t, a_t, r, \widehat{M})p(r|s_t, a_t)dr\right.$$
$$\left. + \int_{\mathcal{S}}\ell_T(s_t, a_t, \widehat{M})p(s_t'|s_t, a_t)ds_t'\right)p_{\mu,M}(s_t, a_t)ds_t$$

$$\widehat{R}_\mu(\widehat{M}) = \sum_{t=0}^{H-1}\int_{\mathcal{S}}\sum_{a_t}\left(\int_{\mathcal{R}}\ell_r(s_t, a_t, r, \widehat{M})p(r|s_t, a_t)dr\right.$$
$$\left. + \int_{\mathcal{S}}\ell_T(s_t, a_t, \widehat{M})p(s_t'|s_t, a_t)ds_t'\right)\widehat{p}_{\mu,M}(s_t, a_t)ds_t$$

$$= \frac{1}{n}\sum_{i=1}^{n}\sum_{t=0}^{H-1}\ell_r(s_t^{(i)}, a_t^{(i)}, r^{(i)}, \widehat{M}) + \ell_T(s_t^{(i)}, a_t^{(i)}, s_t'^{(i)}, \widehat{M}),$$

*where n is the number of trajectories and $s_t^{(i)}$ is the state of the tth step in the ith trajectory. Similarly we define $R_{\pi,u}$ and $\widehat{R}_{\pi,u}$:*

$$R_{\pi,u}(\widehat{M}) = \sum_{t=0}^{H-1}\int_{\mathcal{S}}\sum_{a_t}\frac{\mathbb{1}(a_{0:t}=\pi)}{u_{0:t}}\left(\int_{\mathcal{R}}\ell_r(s_t, a_t, r, \widehat{M})p(r|s_t, a_t)dr\right.$$
$$\left. + \int_{\mathcal{S}}\ell_T(s_t, a_t, \widehat{M})p(s_t'|s_t, a_t)ds_t'\right)p_{M,\mu}(s_t, a_t)ds_t$$

$$\widehat{R}_{\pi,u}(\widehat{M}) = \frac{1}{n}\sum_{i=1}^{n}\sum_{t=0}^{H-1}\frac{\mathbb{1}(a_{0:t}^{(i)}=\pi)}{\widehat{u}_{0:t}}\left[\ell_r(s_t^{(i)}, a_t^{(i)}, r^{(i)}, \widehat{M}) + \ell_T(s_t^{(i)}, a_t^{(i)}, s_t'^{(i)}, \widehat{M})\right],$$

*where $\widehat{u}_{0:t} = \sum_{i=1}^{n}\frac{\mathbb{1}(a_{0:t}^{(i)}=\pi)}{n}$*

We could bound $R_\mu$ and $R_{\pi,u}$ and pseudo-dimension which is a complexity term of the model class. We will use the learning bound about importance sampling and weighted importance sampling in Cortes et al. [4] to bound $R_\mu$ and $R_{\pi,u}$. The following lemma is an immediate consequence of Corollary 2 and section 6 in Cortes et al. [4].

**Lemma 6.** *For a hypothesis class $\mathcal{H}$ over input space $\mathcal{X}$, let $d$ be the pseudo-dimension of a real valued loss function class $\{\ell_h(x), h \in \mathcal{H}, x \in \mathcal{X}\}$. $w(x)$ is a weighting function such that $\mathbb{E}_p[w(x)] = 1$. Let $\widehat{p}$ be the empirical distribution over n samples, and $\widehat{w}(x_i) = nw(x_i)/\sum_{i=1}^{n}w(x_i)$ is the normalized weights. For any $\ell$ in the loss function class. with probability $1-\delta$*

$$|\mathbb{E}_p[w(x)\ell(x)] - \mathbb{E}_{\widehat{p}}[\widehat{w}(x)\ell(x)]| \leq \mathbb{V}_{p,\widehat{p}}[w, \ell]\frac{\mathcal{C}_{n,\delta}^{\mathcal{M}}}{n^{3/8}} + \ell_{\max}\mathbb{V}_{p,\widehat{p}}[w, 1]\frac{\mathcal{C}_{n,\delta}^{\mathcal{M}}}{n^{3/8}} \quad (54)$$

*where* $\mathcal{C}_{n,\delta}^{\mathcal{H}} = 2^{5/4} \left( d \log(2ne/d) + \log(4/\delta) \right)^{3/8}$, $\mathbb{V}_{p,\widehat{p}}[w,\ell] = \max\{\sqrt{\mathbb{E}[w(x)^2 \ell(x)^2]}, \sqrt{\widehat{\mathbb{E}}[w(x)^2 \ell(x)^2]}\}$, *and* $\ell_{\max} = \max_x |\ell(x)|$.

*Proof.* We can decompose the gap into two parts by adding $\mathbb{E}_{\widehat{p}}[w(x)\ell(x)]$:

$$|\mathbb{E}_p[w(x)\ell(x)] - \mathbb{E}_{\widehat{p}}[\widehat{w}(x)\ell(x)]|$$
$$\leq |\mathbb{E}_p[w(x)\ell(x)] - \mathbb{E}_{\widehat{p}}[w(x)\ell(x)]| + |\mathbb{E}_{\widehat{p}}[w(x)\ell(x)] - \mathbb{E}_{\widehat{p}}[\widehat{w}(x)\ell(x)]| \quad (55)$$

For the first part, we can bound it by Corollary 2 from Cortes et al. [4]:

$$|\mathbb{E}_p[w(x)\ell(x)] - \mathbb{E}_{\widehat{p}}[w(x)\ell(x)]| \leq \mathbb{V}_{p,\widehat{p}}[w,\ell]\frac{\mathcal{C}_{n,\delta}^{\mathcal{M}}}{n^{3/8}} \quad (56)$$

For the second part, according to section 6 from Cortes et al. [4], we have that [2]:

$$\left| \frac{1}{n} \left( \widehat{w}(x_i) - w(x_i) \right) \right| = \frac{w(x_i)}{W} \left| 1 - \frac{W}{n} \right| \leq \frac{w(x_i)}{W} \mathbb{V}_{p,\widehat{p}}[w,1]\frac{\mathcal{C}_{n,\delta}^{\mathcal{M}}}{n^{3/8}} \quad (57)$$

where $W = \sum_i w(x_i)$. Then:

$$|\mathbb{E}_{\widehat{p}}[(w(x) - \widehat{w}(x))\ell(x)]| \leq \ell_{\max}\mathbb{E}_p[|w(x) - \widehat{w}(x)|] \quad (58)$$
$$= \ell_{\max} \left[ \frac{1}{n} \sum_i |w(x_i) - \widehat{w}(x_i)| \right] \quad (59)$$
$$\leq \ell_{\max}\mathbb{V}_{p,\widehat{p}}[w,1]\frac{\mathcal{C}_{n,\delta}^{\mathcal{M}}}{n^{3/8}} \quad (60)$$

Thus we finished the proof. $\square$

We will apply this lemma to the risk at each time step $t$ separately. Since $\mathbb{E}_{M,\mu}[\frac{\mathbb{1}(a_{0:t}^{(i)}=\pi)}{u_{0:t}}] = 1$ for each $t$, we can let $w(x) = \frac{\mathbb{1}(a_{0:t}^{(i)}=\pi)}{u_{0:t}}$. In that case $\frac{\mathbb{1}(a_{0:t}^{(i)}=\pi)}{\widehat{u}_{0:t}}$ is the normalized weights $\widehat{w}(x)$. We can also bound of $R_\mu$ from this as well, by setting the weight function to be one. In that case $w = \widehat{w} = 1$ and $\mathbb{V}_{p,\widehat{p}}[1,\ell] \leq \ell_{\max} \leq R_{\max}^2 + V_{\max,t}^2$ for the $t$th step loss function $\ell$.

For the IPM term, using norm-1 reproducing kernel Hilbert space (RKHS) function class for $G$ leads to IPM being the maximum mean discrepancy (MMD) distance. We can bound the gap between MMD distance and its empirical estimation using the following lemma in Sriperumbudur et al.'s work [20]. There are many other choice such as of 1-Lipschitz functions, leading to Wasserstein distance, and $l_\infty$ norm unit ball, leading to total variation distance. There are similar results with those function class and distance measure, with worse bounds. We also use norm-1 RKHS functions and MMD metric in our experiment section.

**Lemma 7.** *(Theorem 11 from Sriperumbudur et al. [20]) Let $\mathcal{X}$ be a measurable space. Suppose $k$ is measurable kernel such that $\sup_{x \in \mathcal{X}} k(x,x) \leq C \leq \infty$ and $\mathcal{H}$ the reproducing kernel Hilbert space induced by $k$. Let $\mathcal{F} = \{f : \|f\|_{\mathcal{H}} = 1\}$, and $\nu = \sup_{x \in \mathcal{X}, f \in \mathcal{F}} |f(x)| < \infty$. Then, with $\widehat{p}$, $\widehat{q}$ the empirical distributions of $p$, $q$ from $m_1$ and $m_2$ samples respectively, and with probability at least 1-$\delta$,*

$$|IPM_{\mathcal{F}}(p,q) - IPM_{\mathcal{F}}(\widehat{p},\widehat{q})| \leq \sqrt{18\nu^2 \ln(4/\delta)C} \left( \frac{1}{\sqrt{m_1}} + \frac{1}{\sqrt{m_2}} \right) \quad (61)$$

**Theorem 2.** *(Restated) Suppose $\mathcal{M}_\Phi$ is a model class of model MDP models based on twice-differentiable, invertible state representation $\phi$'s: $\widehat{M}_\phi = \langle \widehat{r}(s,a), \widehat{T}(s',s,a) \rangle = \langle h_r(\phi(s),a), h_T(\phi(s'),\phi(s),a) \rangle$. Given $n$ H-step trajectories sampled from policy $\mu$, let the loss function for $(s_t, a_t)$ pair at $t^{th}$ step be $\ell_t(s_t, a_t, \widehat{M}_\phi) = \ell_r(s_t, a_t, r_t, \widehat{M}) + \ell_T(s_t, a_t, s_t', \widehat{M})$. Let $d_t$ be the pseudo-dimension of function class $\{\ell_t(s_t, a_t, \widehat{M}_\phi), \widehat{M}_\phi \in \mathcal{M}_\Phi\}$. Suppose $\mathcal{H}$ the reproducing kernel Hilbert space induced by $k$ such that $\sup_{z \in \mathcal{Z}} k(z,z) \leq C \leq \infty$, and*

$\mathcal{F} = \{f : \|f\|_{\mathcal{H}} = 1\}$, and $\nu = \sup_{z \in \mathcal{X}, f \in \mathcal{F}} |f(z)| < \infty$. Assume there exist a constant $B_{\phi,t}$ such that $\frac{1}{B_{\phi,t}} \ell_t(\psi(z), \pi(\psi(z)), \widehat{M}_\phi) \in \mathcal{F}$. Then with probability $1 - 3\delta$, for any $\widehat{M} \in \mathcal{M}_\Phi$:

$$\mathbb{E}_{s_0} \left[ V_{\widehat{M}}^\pi(s_0) - V_M^\pi(s_0) | \widehat{M} \right]^2 \leq MSE_\mu + MSE_\pi \leq 2H\widehat{R}_\mu(\widehat{M}) + 2H\widehat{R}_{\pi,u}(\widehat{M})$$

$$+ 2H \sum_{t=0}^{H-1} B_{\phi,t} \left( IPM_{\mathcal{F}} \left( \widehat{p}_{M,\mu}^{\phi,F}(z_t), \widehat{p}_{M,\mu}^{\phi,CF}(z_t) \right) + \min \left\{ \mathcal{D}_\delta^{\mathcal{F}} \left( \frac{1}{\sqrt{m_{t,1}}} + \frac{1}{\sqrt{m_{t,2}}} \right), 2\nu \right\} \right)$$

$$+ 2H \sum_{t=0}^{H-1} \frac{\mathcal{C}_{n,\delta,t}^{\mathcal{M}}}{n^{3/8}} \left( \mathbb{V}_{p,\hat{p}}[\frac{\mathbb{1}(a_{0:t} = \pi)}{\widehat{u}_{0:t}}, \ell_t] + \mathbb{V}_{p,\hat{p}}[1, \ell_t] + \ell_{t,\max} \mathbb{V}_{p,\hat{p}}[\frac{\mathbb{1}(a_{0:t} = \pi)}{u_{0:t}}, 1] \right)$$

$m_{t,1}$ and $m_{t,2}$ are the number of samples that used to estimate $\widehat{p}_{M,\mu}^{\phi,F}(z_t)$ and $\widehat{p}_{M,\mu}^{\phi,CF}(z_t)$ respectively. $\mathcal{D}_\delta^{\mathcal{F}} = \sqrt{18\nu^2 \ln(4/\delta)C}$. $\mathcal{C}_{n,\delta,t}^{\mathcal{M}} = 2^{5/4} \left( d_t \log(2ne/d_t) + \log(4/\delta) \right)^{3/8}$. $\mathbb{V}_{p,\hat{p}}[w, \ell_t] = \max\{\sqrt{\mathbb{E}_{p_{M,\mu}}[w(s_t, a_t)^2 \ell_t(s_t, a_t)^2]}, \sqrt{\mathbb{E}_{\widehat{p}_{M,\mu}}[w(s_t, a_t)^2 \ell_t(s_t, a_t)^2]}\}$. $\ell_{t,\max} = \max_{s_t,a_t} |\ell_t(s_t, a_t)| \leq R_{\max}^2 + V_{\max,t}^2$.

*Proof.* Applying Lemma 5 to the result in Corollary 1 and plugging the definition of $R_\mu$ and $R_{\pi,u}$ in, we have that:

$$\mathbb{E}_{s_0} \left[ V_{\widehat{M}}^\pi(s_0) - V_M^\pi(s_0) \right]^2$$

$$= 2H \left( \sum_{t=0}^{H-1} B_{\phi,t} IPM_{\mathcal{F}} \left( p_{M,\mu}^{\phi,F}(z_t), p_{M,\mu}^{\phi,CF}(z_t) \right) + R_\mu(\widehat{M}) + R_{\pi,u}(\widehat{M}) - \sigma \right) \quad (62)$$

$$\leq 2H \left( \sum_{t=0}^{H-1} B_{\phi,t} IPM_{\mathcal{F}} \left( p_{M,\mu}^{\phi,F}(z_t), p_{M,\mu}^{\phi,CF}(z_t) \right) + R_\mu(\widehat{M}) + R_{\pi,u}(\widehat{M}) \right) \quad (63)$$

where $\sigma$ is $\sum_{t=0}^{H-1} \sigma_{p_{M,\mu},t} + \sigma_{p_{M,\mu}(\cdot|a_{0:t}=\pi),t} \geq 0$. We will work term by term. First, we can use Lemma 7 for the IPM term:

$$IPM_{\mathcal{F}} \left( p_{M,\mu}^{\phi,F}(z_t), p_{M,\mu}^{\phi,CF}(z_t) \right) \leq IPM_{\mathcal{F}} \left( \widehat{p}_{M,\mu}^{\phi,F}(z_t), \widehat{p}_{M,\mu}^{\phi,CF}(z_t) \right) + \mathcal{D}_\delta^{\mathcal{F}} \left( \frac{1}{\sqrt{m_{t,1}}} + \frac{1}{\sqrt{m_{t,2}}} \right) \quad (64)$$

At the same time, we know that for any two distribution $p, q, 0 \leq IPM_{\mathcal{F}}(p, q) \leq 2\nu$. So:

$$IPM_{\mathcal{F}} \left( p_{M,\mu}^{\phi,F}(z_t), p_{M,\mu}^{\phi,CF}(z_t) \right)$$

$$\leq IPM_{\mathcal{F}} \left( \widehat{p}_{M,\mu}^{\phi,F}(z_t), \widehat{p}_{M,\mu}^{\phi,CF}(z_t) \right) + \min \left\{ \mathcal{D}_\delta^{\mathcal{F}} \left( \frac{1}{\sqrt{m_{t,1}}} + \frac{1}{\sqrt{m_{t,2}}} \right), 2\nu \right\} \quad (65)$$

For $R_\mu$, if we plug $w(s, a) = \widehat{w}(s, a) = 1$, $\ell = \ell_t(s_t, a_t, \widehat{M})$, and $p = p_{M,\mu}(s_t, a_t)$ into Lemma 6, we have that:

$$R_\mu = \sum_{t=0}^{H-1} \mathbb{E}_p[\ell_t(s_t, a_t)] \leq \sum_{t=0}^{H-1} \left( \mathbb{E}_{\widehat{p}}[\ell_t(s_t, a_t)] + \frac{\mathcal{C}_{n,\delta,t}^{\mathcal{M}}}{n^{3/8}} \mathbb{V}_{p,\widehat{p}}[1, \ell_t] \right) \quad (66)$$

$$= \widehat{R}_\mu + \sum_{t=0}^{H-1} \frac{\mathcal{C}_{n,\delta,t}^{\mathcal{M}}}{n^{3/8}} \mathbb{V}_{p,\widehat{p}}[1, \ell_t] \quad (67)$$

An analogous argument can be made for $R_{\pi,u}$ by letting $w(s_t, a_t) = \frac{\mathbb{1}(a_{0:t}=\pi)}{u_{0:t}}$ which leads to that $\widehat{w}(s_t, a_t) = \frac{\mathbb{1}(a_{0:t}=\pi)}{\widehat{u}_{0:t}}$:

$$
\begin{aligned}
R_{\pi,u} &= \sum_{t=0}^{H-1} \mathbb{E}_p[w(s_t, a_t)\ell_t(s_t, a_t)] && (68) \\
&\leq \sum_{t=0}^{H-1} \left( \mathbb{E}_{\widehat{p}}[\widehat{w}(s_t, a_t)\ell_t(s_t, a_t)] + \mathbb{V}_{p,\widehat{p}}[\widehat{w}, \ell_t]\frac{\mathcal{C}_{n,\delta}^{\mathcal{M}}}{n^{3/8}} + \ell_{t,\max}\mathbb{V}_{p,\widehat{p}}[w, 1]\frac{\mathcal{C}_{n,\delta}^{\mathcal{M}}}{n^{3/8}} \right) && (69) \\
&= \widehat{R}_{\pi,u} + \sum_{t=0}^{H-1} \left( \mathbb{V}_{p,\widehat{p}}[\widehat{w}, \ell_t]\frac{\mathcal{C}_{n,\delta}^{\mathcal{M}}}{n^{3/8}} + \ell_{t,\max}\mathbb{V}_{p,\widehat{p}}[w, 1]\frac{\mathcal{C}_{n,\delta}^{\mathcal{M}}}{n^{3/8}} \right) && (70)
\end{aligned}
$$

Thus we finish the proof by combining IPM terms, $R_\mu$ and $R_{\pi,u}$ together. $\qquad\square$

## C.3 IS weights and marginal action probability ratio

**Theorem 3.** *For any evaluation policy $\pi$ and behavior policy $\mu$ satisfying that the support set of $\mu$ covers the support set of $\pi$, we have that the variance of importance sampling weights is no less than the variance of marginal action probability ratios:*

$$
\begin{aligned}
\mathrm{Var}_{\mu,M}\left[ \frac{\prod_{t=0}^{H-1} \pi(a_i|s_i)}{\prod_{t=0}^{H-1} \mu(a_i|s_i)} \right] &\geq \mathrm{Var}_{\mu,M}\left[ \frac{p_{\pi,M}(a_{0:H-1})}{p_{\mu,M}(a_{0:H-1})} \right] \\
&= \mathrm{Var}_{\mu,M}\left[ \frac{\int_{\mathcal{S}^H} \prod_{t=0}^{H-1} \pi(a_i|s_i) \prod_{t=0}^{H-1} T(s_i|s_{i-1}, a_{i-1})\mathrm{d}s_{0:t}}{\int_{\mathcal{S}^H} \prod_{t=0}^{H-1} \mu(a_i|s_i) \prod_{t=0}^{H-1} T(s_i|s_{i-1}, a_{i-1})\mathrm{d}s_{0:t}} \right]
\end{aligned}
$$

*where $T(s_0|s_{-1}, a_{-1})$ is defined as the initial distribution $p_0(s_0)$.*

*Proof.*

$$
\mathrm{Var}_{\mu,M}\left[ \frac{\prod_{t=0}^{H-1} \pi(a_i|s_i)}{\prod_{t=0}^{H-1} \mu(a_i|s_i)} \right] = \mathbb{E}_{\mu,M}\left[ \left( \frac{\prod_{t=0}^{H-1} \pi(a_i|s_i)}{\prod_{t=0}^{H-1} \mu(a_i|s_i)} \right)^2 \right] - \left[ \mathbb{E}_{\mu,M}\left( \frac{\prod_{t=0}^{H-1} \pi(a_i|s_i)}{\prod_{t=0}^{H-1} \mu(a_i|s_i)} \right) \right]^2 \tag{71}
$$

$$
\mathrm{Var}_{\mu,M}\left[ \frac{p_{\pi,M}(a_{0:H-1})}{p_{\mu,M}(a_{0:H-1})} \right] = \mathbb{E}_{\mu,M}\left[ \left( \frac{p_{\pi,M}(a_{0:H-1})}{p_{\mu,M}(a_{0:H-1})} \right)^2 \right] - \left[ \mathbb{E}_{\mu,M}\left( \frac{p_{\pi,M}(a_{0:H-1})}{p_{\mu,M}(a_{0:H-1})} \right) \right]^2 \tag{72}
$$

Among them, the expectation of marginal action probability ratio is equal to the expectation of IS weights:

$$\mathbb{E}_{\mu,M}\left(\frac{p_{\pi,M}(a_{0:H-1})}{p_{\mu,M}(a_{0:H-1})}\right) \tag{73}$$

$$= \int_{\mathcal{S}^H} \sum_{a_0,\ldots,a_{H-1}} \prod_{t=0}^{H-1} \mu(a_i|s_i) \prod_{t=0}^{H-1} T(s_i|s_{i-1},a_{i-1}) \frac{p_{\pi,M}(a_{0:H-1})}{p_{\mu,M}(a_{0:H-1})} \mathrm{d}s_{0:t} \tag{74}$$

$$= \sum_{a_0,\ldots,a_{H-1}} \left(\int_{\mathcal{S}^H} \prod_{t=0}^{H-1} \mu(a_i|s_i) \prod_{t=0}^{H-1} T(s_i|s_{i-1},a_{i-1})\mathrm{d}s_{0:t}\right) \frac{p_{\pi,M}(a_{0:H-1})}{p_{\mu,M}(a_{0:H-1})} \tag{75}$$

$$= \sum_{a_0,\ldots,a_{H-1}} p_{\mu,M}(a_{0:H-1}) \frac{p_{\pi,M}(a_{0:H-1})}{p_{\mu,M}(a_{0:H-1})} \tag{76}$$

$$= \sum_{a_0,\ldots,a_{H-1}} p_{\pi,M}(a_{0:H-1}) = 1 \tag{77}$$

$$\mathbb{E}_{\mu,M}\left(\frac{\prod_{t=0}^{H-1}\pi(a_i|s_i)}{\prod_{t=0}^{H-1}\mu(a_i|s_i)}\right) \tag{78}$$

$$= \int_{\mathcal{S}^H} \sum_{a_0,\ldots,a_{H-1}} \prod_{t=0}^{H-1} \mu(a_i|s_i) \prod_{t=0}^{H-1} T(s_i|s_{i-1},a_{i-1}) \left(\frac{\prod_{t=0}^{H-1}\pi(a_i|s_i)}{\prod_{t=0}^{H-1}\mu(a_i|s_i)}\right) \mathrm{d}s_{0:t} \tag{79}$$

$$= \sum_{a_0,\ldots,a_{H-1}} \int_{\mathcal{S}^H} \prod_{t=0}^{H-1} T(s_i|s_{i-1},a_{i-1}) \prod_{t=0}^{H-1} \pi(a_i|s_i)\mathrm{d}s_{0:t} \tag{80}$$

$$= \sum_{a_0,\ldots,a_{H-1}} p_{\pi,M}(a_{0:H-1}) = 1 \tag{81}$$

Thus the second term in Equation 71 and Equation 72 are the same. Now we are going to prove that

$$\mathbb{E}_{\mu,M}\left[\left(\frac{\prod_{t=0}^{H-1}\pi(a_i|s_i)}{\prod_{t=0}^{H-1}\mu(a_i|s_i)}\right)^2\right] \geq \mathbb{E}_{\mu,M}\left[\left(\frac{p_{\pi,M}(a_{0:H-1})}{p_{\mu,M}(a_{0:H-1})}\right)^2\right] \tag{82}$$

$$\mathbb{E}_{\mu,M}\left[\left(\frac{\prod_{t=0}^{H-1}\pi(a_i|s_i)}{\prod_{t=0}^{H-1}\mu(a_i|s_i)}\right)^2\right] \tag{83}$$

$$= \int_{\mathcal{S}^H} \sum_{a_0,\ldots,a_{H-1}} \prod_{t=0}^{H-1} \mu(a_i|s_i) \prod_{t=0}^{H-1} T(s_i|s_{i-1},a_{i-1}) \left(\frac{\prod_{t=0}^{H-1}\pi(a_i|s_i)}{\prod_{t=0}^{H-1}\mu(a_i|s_i)}\right)^2 \mathrm{d}s_{0:t} \tag{84}$$

$$= \sum_{a_0,\ldots,a_{H-1}} \int_{\mathcal{S}^H} \prod_{t=0}^{H-1} T(s_i|s_{i-1},a_{i-1}) \frac{\left(\prod_{t=0}^{H-1}\pi(a_i|s_i)\right)^2}{\prod_{t=0}^{H-1}\mu(a_i|s_i)} \mathrm{d}s_{0:t} \tag{85}$$

$$= \sum_{a_0,\ldots,a_{H-1}} \int_{\mathcal{S}^H} \frac{\left(\prod_{t=0}^{H-1}T(s_i|s_{i-1},a_{i-1})\prod_{t=0}^{H-1}\pi(a_i|s_i)\right)^2}{\prod_{t=0}^{H-1}T(s_i|s_{i-1},a_{i-1})\prod_{t=0}^{H-1}\mu(a_i|s_i)} \mathrm{d}s_{0:t} \tag{86}$$

$$\mathbb{E}_{\mu,M}\left[\left(\frac{p_{\pi,M}(a_{0:H-1})}{p_{\mu,M}(a_{0:H-1})}\right)^2\right] \tag{87}$$

$$= \int_{\mathcal{S}^H} \sum_{a_0,\ldots,a_{H-1}} \prod_{t=0}^{H-1} \mu(a_i|s_i) \prod_{t=0}^{H-1} T(s_i|s_{i-1},a_{i-1}) \left(\frac{p_{\pi,M}(a_{0:H-1})}{p_{\mu,M}(a_{0:H-1})}\right)^2 \mathrm{d}s_{0:t} \tag{88}$$

$$= \sum_{a_0,\ldots,a_{H-1}} \left(\int_{\mathcal{S}^H} \prod_{t=0}^{H-1} \mu(a_i|s_i) \prod_{t=0}^{H-1} T(s_i|s_{i-1},a_{i-1})\mathrm{d}s_{0:t}\right) \left(\frac{p_{\pi,M}(a_{0:H-1})}{p_{\mu,M}(a_{0:H-1})}\right)^2 \tag{89}$$

$$= \sum_{a_0,\ldots,a_{H-1}} p_{\mu,M}(a_{0:H-1}) \left(\frac{p_{\pi,M}(a_{0:H-1})}{p_{\mu,M}(a_{0:H-1})}\right)^2 \tag{90}$$

$$= \sum_{a_0,\ldots,a_{H-1}} \frac{(p_{\pi,M}(a_{0:H-1}))^2}{p_{\mu,M}(a_{0:H-1})} \tag{91}$$

$$= \sum_{a_0,\ldots,a_{H-1}} \frac{\left(\int_{\mathcal{S}^H} \prod_{t=0}^{H-1} \pi(a_i|s_i) \prod_{t=0}^{H-1} T(s_i|s_{i-1},a_{i-1})\mathrm{d}s_{0:t}\right)^2}{\int_{\mathcal{S}^H} \prod_{t=0}^{H-1} \mu(a_i|s_i) \prod_{t=0}^{H-1} T(s_i|s_{i-1},a_{i-1})\mathrm{d}s_{0:t}} \tag{92}$$

Now we only need to prove that for any $a_0, a_1, \ldots a_{H-1}$:

$$\int_{\mathcal{S}^H} \frac{\left(\prod_{t=0}^{H-1} T(s_i|s_{i-1},a_{i-1}) \prod_{t=0}^{H-1} \pi(a_i|s_i)\right)^2}{\prod_{t=0}^{H-1} T(s_i|s_{i-1},a_{i-1}) \prod_{t=0}^{H-1} \mu(a_i|s_i)} \mathrm{d}s_{0:t} \tag{93}$$

$$\geq \frac{\left(\int_{\mathcal{S}^H} \prod_{t=0}^{H-1} \pi(a_i|s_i) \prod_{t=0}^{H-1} T(s_i|s_{i-1},a_{i-1})\mathrm{d}s_{0:t}\right)^2}{\int_{\mathcal{S}^H} \prod_{t=0}^{H-1} \mu(a_i|s_i) \prod_{t=0}^{H-1} T(s_i|s_{i-1},a_{i-1})\mathrm{d}s_{0:t}} \tag{94}$$

Multiplying both sides $\int_{\mathcal{S}^H} \prod_{t=0}^{H-1} \mu(a_i|s_i) \prod_{t=0}^{H-1} T(s_i|s_{i-1},a_{i-1})\mathrm{d}s_{0:t}$ we have that:

$$\int_{\mathcal{S}^H} \prod_{t=0}^{H-1} \mu(a_i|s_i) \prod_{t=0}^{H-1} T(s_i|s_{i-1},a_{i-1})\mathrm{d}s_{0:t} \int_{\mathcal{S}^H} \frac{\left(\prod_{t=0}^{H-1} T(s_i|s_{i-1},a_{i-1}) \prod_{t=0}^{H-1} \pi(a_i|s_i)\right)^2}{\prod_{t=0}^{H-1} T(s_i|s_{i-1},a_{i-1}) \prod_{t=0}^{H-1} \mu(a_i|s_i)}\mathrm{d}s_{0:t} \tag{95}$$

$$\geq \left(\int_{\mathcal{S}^H} \prod_{t=0}^{H-1} \pi(a_i|s_i) \prod_{t=0}^{H-1} T(s_i|s_{i-1},a_{i-1})\mathrm{d}s_{0:t}\right)^2 \tag{96}$$

This inequality holds by applying Cauchy-Schwarz inequality. Thus we finish the proof. $\qquad\square$

## D  Proof of Section 5

**Corollary 2.** *Let $\widehat{M}^*_{\phi^*} = \arg\min_{\widehat{M}_\phi} \mathcal{L}(\widehat{M}_\phi; \alpha_t)$ for a large enough $\alpha$ such that $\alpha_t > B_{\phi^*,t}$. Under the same definition and assumption in Theorem 2, we have that:*

$$\mathbb{E}_{s_0}\left[V^\pi_{\widehat{M}^*_{\phi^*}}(s_0) - V^\pi_M(s_0)\right]^2 \leq O\left(\frac{1}{n^{3/8}}\right) + \sum_{t=1}^{H-1} O\left(\frac{1}{\sqrt{m_{t,1}}} + \frac{1}{\sqrt{m_{t,2}}}\right)$$

$$+ 2H \min_{\widehat{M}_\phi \in \mathcal{M}} \left(R_\mu(\widehat{M}_\phi) + R_{\pi,u}(\widehat{M}_\phi) + \sum_{t=0}^{H-1} \alpha_t IPM_{\mathcal{F}}\left(p^{\phi,F}_{M,\mu}(z_t), p^{\phi,CF}_{M,\mu}(z_t)\right)\right)$$

*Proof.* Let $M^*_{\phi_0^*}$ be the model that minimizes the expected risk and IPM term:

$$M^*_{\phi_0^*} = \arg\min_{\widehat{M}_\phi} \left(R_\mu(\widehat{M}_\phi) + R_{\pi,u}(\widehat{M}_\phi) + \sum_{t=0}^{H-1} \alpha_t IPM_{\mathcal{F}}\left(p^{\phi,F}_{M,\mu}(z_t), p^{\phi,CF}_{M,\mu}(z_t)\right)\right)$$

From theorem 2 we have that:

$$\mathbb{E}_{s_0}\left[V^{\pi}_{\widehat{M}^*_{\phi^*}}(s_0) - V^{\pi}_M(s_0)\right]^2$$

$$\leq O\left(\frac{1}{n^{3/8}}\right) + \sum_{t=1}^{H-1} O\left(\frac{1}{\sqrt{m_{t,1}}} + \frac{1}{\sqrt{m_{t,2}}}\right) + 2HL(\widehat{M}^*_{\phi^*}; B_{\phi^*,t}) \quad (97)$$

$$\leq O\left(\frac{1}{n^{3/8}}\right) + \sum_{t=1}^{H-1} O\left(\frac{1}{\sqrt{m_{t,1}}} + \frac{1}{\sqrt{m_{t,2}}}\right) + 2HL(\widehat{M}^*_{\phi^*}; \alpha_t) \quad (98)$$

$$\leq O\left(\frac{1}{n^{3/8}}\right) + \sum_{t=1}^{H-1} O\left(\frac{1}{\sqrt{m_{t,1}}} + \frac{1}{\sqrt{m_{t,2}}}\right) + 2H\mathcal{L}(M^*_{\phi_0^*}; \alpha_t) \quad (99)$$

The first step follows from Theorem 2. The second step is from the fact that $\alpha_t > B_{\phi^*,t}$. The third step is from that $\widehat{M}^*_{\phi^*} = \arg\min_{\widehat{M}_\phi} \mathcal{L}(\widehat{M}_\phi; \alpha_t)$. Then we can bound the empirical loss term $\mathcal{L}(M^*_{\phi_0^*}; \alpha_t)$ by the expected value of it:

$$\mathcal{L}(M^*_{\phi_0^*}; \alpha_t) = \widehat{R}_\mu(M^*_{\phi_0^*}) + \widehat{R}_{\pi,u}(M^*_{\phi_0^*}) + \sum_{t=0}^{H-1} \alpha_t \mathrm{IPM}_{\mathcal{F}}\left(\widehat{p}^{\phi_0^*,F}_{M,\mu}(z_t), \widehat{p}^{\phi_0^*,CF}_{M,\mu}(z_t)\right)$$

$$+ \frac{\mathfrak{R}(\widehat{M}_\phi)}{n^{3/8}} \quad (100)$$

$$\leq R_\mu(M^*_{\phi_0^*}) + R_{\pi,u}(M^*_{\phi_0^*}) + \sum_{t=0}^{H-1} \alpha_t \mathrm{IPM}_{\mathcal{F}}\left(p^{\phi_0^*,F}_{M,\mu}(z_t), p^{\phi_0^*,CF}_{M,\mu}(z_t)\right)$$

$$+ O\left(\frac{1}{n^{3/8}}\right) + O\left(\frac{1}{\sqrt{m_{t,1}}} + \frac{1}{\sqrt{m_{t,2}}}\right) \quad (101)$$

This follows from using Lemma 6 and Lemma 7 similarly with Equation 65, 67, 70 but in different direction, together with the fact that $\frac{\mathfrak{R}(\widehat{M}_\phi)}{n^{3/8}} = O\left(\frac{1}{n^{3/8}}\right)$.

Put this into equation 99, we have that

$$\mathbb{E}_{s_0}\left[V^{\pi}_{\widehat{M}^*_{\phi^*}}(s_0) - V^{\pi}_M(s_0)\right]^2 \leq 2H\left(R_\mu(M^*_{\phi_0^*}) + R_{\pi,u}(M^*_{\phi_0^*})\right.$$

$$+ \sum_{t=0}^{H-1} \alpha_t \mathrm{IPM}_{\mathcal{F}}\left(p^{\phi_0^*,F}_{M,\mu}(z_t), p^{\phi_0^*,CF}_{M,\mu}(z_t)\right)\right) + O\left(\frac{1}{n^{3/8}}\right) + \sum_{t=1}^{H-1} O\left(\frac{1}{\sqrt{m_{t,1}}} + \frac{1}{\sqrt{m_{t,2}}}\right) \quad (102)$$

Thus we finished the proof. $\qquad \square$

Under assumption about support set of $\mu$, $m_{t,1}, m_{t,2} \to \infty$ when $n \to \infty$. Then an immediate consequence from this corollary is that, if there exists an MDP and representation model in our model class that could achieve no generalization error,

$$\min_{\widehat{M}_\phi}\left(R_\mu(\widehat{M}_\phi) + R_{\pi,u}(\widehat{M}_\phi) + \sum_{t=0}^{H-1} \alpha_t \mathrm{IPM}_{\mathcal{F}}\left(p^{\phi,F}_{M,\mu}(z_t), p^{\phi,CF}_{M,\mu}(z_t)\right)\right) = 0,$$

then $\lim_{n\to\infty} \mathbb{E}_{s_0}\left[V^{\pi}_{\widehat{M}^*_{\phi^*}}(s_0) - V^{\pi}_M(s_0)\right]^2 \to 0$ and estimator $V^{\pi}_{\widehat{M}^*_{\phi^*}}(s_0)$ is a consistent estimator for any $s_0$.

## E Details of Experiment

We will clarify the details of the Cart Pole and Mountain Car experiment and provide results from additional OPPE methods.

**Details of the domain** For Cart Pole domain, we follow the same settings as in the OpenAI Gym [3] CartPole-v0 environment. The state consists of 4 features: position, speed, angle and angular speed. The agent can take two actions: move to the left or to the right. The trajectory will end either when the time step is larger than 200 or when the absolute value of position or angle is larger than the threshold. The goal in this domain is to control a cart as long as possible. We will receive the reward after each time step if the cart is under control, and the trajectory ends when the cart falls.

We include two different variants in this domain: long horizon and short horizon. For long horizon, we learn a near-optimal Q function, and use the greedy policy as evaluation policy and $\epsilon$−greedy policy with $\epsilon = 0.2$ as behavior policy. The average value, which is also the average length of trajectories, of the evaluation policy is 195 and the average value of the behavior policy is 190. For shorter horizon, we learn a weaker Q function and generate the policies in the same way, with the average value of 23.8 and 24 respectively. The reason that we learn a near optimal but not optimal policy for the long horizon is that the optimal policy can always hold the cart for 200 steps (max length), which makes it easy to estimate since there is no possibility of overestimating it.

For Mountain Car domain, we follow the same settings as in the OpenAI Gym [3] MountainCar-v0 environment. The state consists of 4 features: position and velocity. The agent can take two actions: accelerate to the left or to the right. The trajectory will end either when the time step is larger than 200 or when the position exceeds the threshold. The goal in this domain is to control the car to reach the top of mountain as soon as possible. We will receive a negative reward after each time step.

**Details of our model** Our model has three parts: a representation module, a reward module, and a transition module. The representation module is a one layer feed-forward network that takes the state as input and outputs a 32-dimension representation. The reward module takes the representation as input and outputs $A = 2$ predictions, corresponding to 2 different actions. The transition module is similar to the reward module, but it predicts the difference between state and next state which is a widely-used trick for transition dynamics modeling. Both the reward module and transition module are feed-forward networks with no hidden layer. We optimize the model using Adam. Since this domain has variable length of trajectories, we also learn the condition of terminal state. The only domain prior we assume is that we know the maximum length of a trajectory is 200.

We also need to explain the details of transition loss $\ell_T$. Since this domain is a deterministic domain, the loss function turns to be:

$$\ell_T(s_t, a_t, s'_t, \widehat{M}) = \left( V^\pi_{\widehat{M}, H-t-1}(s'_t) - V^\pi_{\widehat{M}, H-t-1}(s') \right)^2 , \tag{103}$$

where $s'$ is the predicted next state prediction and $s'_t$ is the logged next state in dataset. Since repeatedly performing planning at training time is very computationally-intensive, it is difficult to get the function $V^\pi_{\widehat{M}, H-t-1}(s)$. It is also challenging to compute the the derivative of this with respect to $s$. If we assume the resulting value is $L$-Lipschitz, then this loss can be bounded by $L(s' - s'_t)^2$. This is slightly different to the algorithmic part in the main body but it will still be an upper bound of $\ell_T$ in the main body. In this experiment we set $L = 1$.

If we are in a discrete state space, the transition loss $\ell_T$ turns to be:

$$\ell_T(s_t, a_t, s'_t, \widehat{M}) = \sum_{s' \in \mathcal{S}} V^\pi_{\widehat{M}, H-t-1}(s') \left( \widehat{T}(s'|s,a) - \mathbb{1}(s'_t - s') \right)^2 \tag{104}$$

We can use similar trick with double Q learning for DQN: doing value iteration to generate a target value vector $V^\pi_{\widehat{M}, H-t-1}(s')$, and view this as constant vector when we compute derivative with $s'$. Then this loss becomes a weighted MSE loss. We can update the target value vector $V^\pi_{\widehat{M}, H-t-1}(s')$ every several episodes.

**Methods** We compare several different methods: **1) RepBM** The proposed method. **2) AM** We compare our method $RepBM$ with a baseline approximate model, which uses the exactly same model class as our model, with the objective of minimizing the on-policy loss $R_\mu$. This is a straight-forward way to fit a regression model without any off-policy adjustment. **3) MRDR** we also compare with the more robust doubly robust (MRDR) method, which proposed a new way to train a Q function and use it in doubly robust. MRDR trains the Q function to minimize:

$$\frac{1}{n} \sum_{i=1}^n \sum_{t=0}^{H-1} (w_{0:t}^{(i)})^2 \frac{1 - \mu(a_t^{(i)}|s_t^{(i)})}{\mu(a_t^{(i)}|s_t^{(i)})} \left( \bar{R}_{t:H-1}^{(i)} - \widehat{Q}^\pi(s_t^{(i)}, a_t^{(i)}) \right), \tag{105}$$

where $\bar{R}_{t:H-1}^{(i)} = \sum_{j=t+1}^{H-1} w_{t+1:j}^{(i)} r_j^{(i)}$ is the per-decision IS return from $t+1$ to $H-1$, and $w$'s are IS weights. **4) MRDR-WIS** Since this objective function can be very noisy and hard to fit when IS weights are high-variance, we also test another variant of MRDR by changing $\bar{R}_{t:H-1}^{(i)}$ to a weighted per-decision IS return from $t+1$ to $H-1$.

Within each one of the methods above, we test five different kinds of estimator. We have a pure MDP/Q model estimator, doubly robust (DR) using that MDP/Q model and weighted doubly robust (WDR) using that MDP/Q model. We evaluate a deterministic evaluation policy, which will result in most of the IS weights being zero; once an IS weight at one timestep is zero, then the product of all IS weights after that step will be zero. This setting is challenging for importance sampling and DR. We also test a very simple idea to avoid this problem – we add a slight noise perturbation into the evaluation policy ($\epsilon = 0.01$), and treat it as the true evaluation policy to generate IS weights for DR and WDR. The additional noise is small enough so that the error introduced by this is negligible compared with the MSEs of estimators. We call these variants of DR and WDR soft DR and soft WDR respectively.

We also compare with importance sampling (IS), weighted IS (WIS), soft IS, soft WIS, per-decision importance sampling (PDIS), weighted PDIS (WPDIS), soft PDIS, soft WPDIS. The soft methods are produced by changing the IS weights using the soft evaluation policy.

We report the results in Table 4 and 5. Note that in the long horizon case, IS weights are all zero so WIS estimator is not defined. Though it is clear that for a single individual in continuous state space, IS and DR would not produce meaningful results due to the fact they only estimate from one trajectory, here we still include the IS and DR estimates for MSE for individual policy values. Not surprisingly we observe that those results are enormous which verifies that plain IS and DR are not reasonable estimators for individual value.

Table 4: Root MSE for Cart Pole

|  | Long horizon | | Short horizon | |
|  | MSE (mean) | MSE (individual) | MSE (mean) | MSE (individual) |
| --- | --- | --- | --- | --- |
| RepBM | **0.412** | **1.033** | 0.078 | **0.481** |
| DR(RepBM) | 1.359 | 40.820 | 0.021 | 0.789 |
| WDR(RepBM) | 0.619 | 17.760 | 0.026 | 0.857 |
| Soft DR(RepBM) | 1.608 | 53.390 | **0.020** | 0.686 |
| Soft WDR(RepBM) | 0.730 | 24.95 | 20.59 | 634.7 |
| AM | 0.754 | 1.313 | 0.125 | 0.551 |
| DR(AM) | 1.786 | 58.66 | 0.024 | 0.863 |
| WDR(AM) | 0.706 | 19.73 | 0.025 | 0.929 |
| Soft DR(AM) | 1.613 | 52.56 | **0.020** | 0.744 |
| Soft WDR(AM) | 0.848 | 29.71 | 20.28 | 640.4 |
| AM ($\pi$) | 41.80 | 47.63 | 0.1233 | 0.5974 |
| MRDR's Q | 151.1 | 151.9 | 3.013 | 3.823 |
| MRDR | 202.0 | 7055 | 0.258 | 8.266 |
| WMRDR | 123.6 | 1049 | 2.343 | 59.640 |
| Soft MRDR | 813.8 | 2590 | 0.211 | 6.758 |
| Soft WMRDR | 92.00 | 2669 | 22.550 | 601.4 |
| MRDR-WIS's Q | 143.9 | 145.1 | 2.486 | 3.440 |
| MRDR-WIS | 190.9 | 6106 | 0.248 | 8.075 |
| WMRDR-WIS | 122.0 | 1054 | 2.599 | 68.60 |
| Soft MRDR-WIS | 746.9 | 23610 | 0.199 | 6.626 |
| Soft WMRDR-WIS | 108.3 | 2992 | 21.26 | 570.6 |
| IS | 194.500 | 194.7 | 2.860 | 93.87 |
| WIS | - | - | 0.505 | 93.86 |
| Soft IS | 187.9 | 1115 | 2.179 | 70.78 |
| Soft WIS | 8.144 | 4698 | 0.380 | 70.55 |
| PSIS | 477.5 | 1526 | 1.083 | 36.67 |
| WPSIS | 125.9 | 622.5 | 1.819 | 63.59 |
| Soft PSIS | 215.2 | 6853 | 0.903 | 30.08 |
| Soft WPSIS | 4.225 | 1983 | 24 | 678.2 |

Table 5: Root MSE for Mountain Car

| | MSE (mean) | MSE (individual) |
|---|---|---|
| RepBM | **12.31** | **31.38** |
| DR(RepBM) | 135.8 | 4352 |
| WDR(RepBM) | 27.27 | 790.7 |
| Soft DR(RepBM) | 59.9 | 1929 |
| Soft WDR(RepBM) | 22.6 | 825.4 |
| AM | 17.15 | 36.36 |
| DR(AM) | 141.6 | 4548 |
| WDR(AM) | 24.89 | 756.9 |
| Soft DR(AM) | 66.45 | 2129 |
| Soft WDR(AM) | 23.79 | 831.7 |
| AM ($\pi$) | 72.61 | 79.46 |
| MRDR's Q | 135.4 | 138.1 |
| MRDR | 172.7 | 5427 |
| WMRDR | 78.34 | 1400 |
| Soft MRDR | 4481 | 125100 |
| Soft WMRDR | 5631 | 104500 |
| MRDR-WIS's Q | 140.5 | 143.1 |
| MRDR-WIS | 212 | 6975 |
| WMRDR-WIS | 110.1 | 2101 |
| Soft MRDR-WIS | 5308 | 139000 |
| Soft WMRDR-WIS | 8564 | 167700 |
| IS | 149.7 | 152.2 |
| WIS | nan | nan |
| Soft IS | 208.5 | 3936 |
| Soft WIS | 301.3 | 3862 |
| PSIS | 108.6 | 2334 |
| WPSIS | 99.79 | 440.7 |
| Soft PSIS | 117.1 | 3597 |
| Soft WPSIS | 45.8 | 1924 |

**Evaluation** Thomas and Brunskill [22] discussed that it is not obvious how to use the trajectories to fairly compare DR, IS and AM estimators, in Appendix D.4 from [22]. There are three ways that are reasonable: the first way is that AM and DR estimators should be provided with additional trajectories that are not available to IS, which are used to learn the model. This can be viewed as the additional domain prior knowledge. This is the setting in MRDR's experiment [7]. The second way is that all methods should have the same amount of data. DR methods should split the data into two parts to learn the model and IS weights separately. That partition keeps the unbiasedness of DR, but reduces the size of available samples for model learning in DR. The third way is that all methods should have the same amount of data. The DR method reuses the data to learn the model and compute IS weights. This helps DR methods to achieve best empirical performance in Thomas and Brunskill [22]. There is not necessarily a "correct" answer to this question. We follow the third setting to make both DR and IS stronger baselines.

We sample 1024 trajectories to generate off-policy estimators. For our method and AM method, we split the data into a training set (90%) and a validation set (10%) and use the validation set to tune the model structure and optimization settings. To compute the MSE of an individual value, we record the initial state of the 1024 trajectories and roll-out from true environment to get the true policy over those initial states as ground truth. We use the average policy value over these initial states as the ground truth for MSE of mean value. We repeat the whole process for $N = 100$ runs and report the square root of averaged MSEs (for both individual and mean).

**Effect of parameter $\alpha$**

We study the effect of the hyper-paramter $\alpha$ in the IPM terms on the estimation results. We show the MSE of RepBM trained using different $\alpha$.

Table 6: Root MSEs of RepBM with different $\alpha$ for the cartpole domain

| Long horizon | $\alpha = 0$ | $\alpha = 0.01$ | $\alpha = 0.1$ | $\alpha = 1$ | $\alpha = 10$ |
|---|---|---|---|---|---|
| Mean | 0.554 | 0.412 | 0.406 | **0.389** | 2.287 |
| Individual | 1.178 | 1.033 | **1.008** | 1.023 | 3.903 |

| Short horizon | $\alpha = 0$ | $\alpha = 0.01$ | $\alpha = 0.1$ | $\alpha = 1$ | $\alpha = 10$ |
|---|---|---|---|---|---|
| Mean | 0.114 | **0.078** | 0.114 | 0.357 | 0.365 |
| Individual | 0.672 | **0.481** | 0.702 | 1.684 | 1.545 |

## E.1 Further discussion

An interesting issue is about the effect of the horizon. Although the "marginal" IS weights have less variance than IS weights, there is still a concern when the horizon is very long and the overlap of the behavior policy and the evaluation policy is small. That also has an effect on the IPM term: we would not have enough factual/counterfactual samples to estimate the IPMs, for large time steps $t$. In that case, the IPMs only effectively adjust the representation for the earlier of the trajectories. Both of the experimental domains actually encounter this case, and the experimental results show that RepBM still outperforms other methods. In the Cart Pole domain it is clear that RepBM can still benefit from IPM.

## Footnotes

[2]Note that the definition of $\widehat{w}(x_i)$ in Cortes et al. [4] is different with ours by a constant $n$. Here $\widehat{w}(x)$ follows our definition.