[Reviews · NeurIPS 2018]

Reviewer 1



This paper deals with the problem of off-policy policy evaluation in MDPs. The paper seems to extend the prior work on the contextual bandit to the multi-step sequential decision-making setting, where the algorithm focuses on learning representation for MDPs by minimizing the upper bound of MSE. The upper bound considers balancing the distribution of state representation when factual and counterfactual action history are given respectively. They provide a finite sample generalization error, and experimental results show that their method outperforms the baselines on Cartpole and Sepsis domains. I enjoyed reading the paper. I think it solves an important problem in an elegant way and does not leave many questions unanswered. - Can the proposed method be extended to the setting when the evaluation policy is stochastic, or to off-policy control setting? - How \mathfrak{R}(M_\phi) of the fourth term in Eq. (10) was defined specifically for the experiments? - Cartpole used in the experiment seems to be a deterministic environment. Does RepBM also work well in the stochastic environment?

Reviewer 2



The paper studies how to fit a model for the purpose of off-policy policy evaluation (OPPE), that is, estimating the value function of a policy using data from a different policy. Instead of using a simple loss function under the data distribution, the proposed method addresses the distribution shift issue by incorporating data distribution under target policy, approximated by importance sampling. Doing so alone, however, results in data sparsity, so the ultimate objective also includes the error of the behavior policy to stabilize and regularize learning. Finite sample generalization error bounds are proved for the proposed approaches, and empirical evaluation compares the approaches to other OPE methods for value function and expected return estimations. I like the research question of how to estimate a model for OPE that takes into account the distribution shift issue. The ideas in the paper make a lot of intuitive sense. The paper is well written and pleasant to read, and the experiments show some promising results. Now let me turn to weaknesses. The first major weakness is that, the theory part does not really support the proposed method well (despite the amount of maths involved). One thing I want to highlight here is that, as far as I understand, the introduction of MSE_mu in the objective is not justified at all by the theorems. Namely, if we get rid of MSE_mu we should be able to obtain *better* upper bounds; see Bullet 4. Neither is it justified empirically. It should be noted that some of the theoretical caveats are addressed by the positive empirical results. That said, the experiment setup on the Sepsis dataset, where the most promising results are shown, is very questionable; see bullet 6 below. Put it bluntly, I would be more comfortable accepting the paper if this section is simply dropped (unless the methodology can be rigorously justified). Overall this is a borderline paper to me, and I temporarily lean very slightly towards rejection. That said, I really like the research topic and find the ideas of the paper quite nice, so I am ready to be convinced by the authors and the other reviewers (esp. on the Sepsis data issue) and change my mind. Detailed comments: 1. Why is E_{s0}[(...)^2] the objective to consider at all? In particular, why do we only consider the initial state distribution of s0, and ignore the later time steps? Details follow: (a) s0 can be deterministic w.l.o.g (just add a fake initial state that transitions to the true initial distribution), in which case this objective is no different from the average value objective (APV, line 43). So for the objective to make a difference, we are at the mercy of the environment---that its initial state distribution is well spread somehow. When is this the case? (b) Which objective is right really depends on what you want to do with the estimated value function. In the experiment section, the best use case of the proposed method is still average value estimation. If this is the ultimate use case, *for the purpose of theoretical development*, you should start with the average value objective, and then explain why using the proposed objective as a *surrogate* actually leads to better average value estimations. (c) Now there is definitely more we can do with a value function other than turning it into a scalar return---for example, policy improvement. But if policy improvement is the goal, we need to consider distributions of states beyond the initial step (see e.g., the CPI paper by Kakade and Langford). 2. Eq.4 is said to be an upper bound on the OPPE error. Isn't OPPE error simply the first term? Calling Eq.4 an upper bound is mathematically correct but perhaps conceptually misleading: for example, we say that hinge-loss is an upper bound on 0-1 loss, but seldom say that the L-2 regularized objective is an upper bound on the empirical risk (note it could upper bound population risk as in SRM and that's legit, but that's not what's going on in this paper). Overall I feel that the introduction of the MSE_mu is more heuristic than it sounds to be. The looseness of Eq.4 also factors into Thm 2. (The first step in the equation between Line 225 and 226 performs this relaxation.) This brings to my next point... 3. One thing that is missing is the comparison to baseline approaches in theory. What guarantees can we give for simply fitting the model on data distribution? Is the guarantee in the current paper better than that of the baseline approach? 4. As far as I understand, the current analyses do not justify the introduction of MSE_mu, since Thm 2 relaxes MSE_pi to MSE_pi + MSE_mu and bounds them separately. If we just drop MSE_mu, we should get better upper bounds? Please comment on this in the rebuttal. 5. The empirical evaluation is thorough and shows promising results. However, given the caveat in bullet 3, I think it's very important to perform an ablation study and see how the method performs without the MSE_mu part. 6. For the Sepsis data, I am not sure why the artificial split of data is valid. For example, if the actions made during data collection depend on the patient's characteristics, such a split of data could give us two subpopulations that are very different in distribution, breaking the basic assumptions of off-policy evaluation. Unfortunately, I cannot find any description of the behavior policy in the paper. More generally speaking, what this section does is to "mimic" an A/B test using observational data. There is extensive literature related to this from Stats/Econ (e.g., instrumental variables), and this usually needs to be done with extreme care. While I am not an expert on this, I feel that the paper's treatment here is surprisingly casual. Also note that it wouldn't be much of a problem if someone does this in the algorithm: if an algorithm does something wrong it will perform poorly, and the poor performance will be observed correctly. The problem is more serious here because the methodology is used to form the groundtruth, hence any mistake would affect the validity of the conclusions we draw from the results. Minor - "Individual" vs "average" policy value: the terminologies here are somewhat unconventional. Perhaps consider "value function" vs "expected return" of a policy. - References need a very careful pass: [5] and [9] should use conference versions instead of arXiv, author name in [8], [20] has "\phi" in it, etc. - Line 197, space before the comma. ================================== After rebuttal: Thanks for the rebuttal. I am happy to learn that the authors agree with quite a few of my concerns and have proposed concrete revisions to address them. Despite that the revisions seem quite substantial, I am reasonably confident that the authors can implement them faithfully for camera-ready, so I am raising my score accordingly. Re the initial state issue, the authors' argument is in line with my 1(b) in review, so you might want to consider the alternative presentation I mentioned there.

Reviewer 3



The paper proposes a new approach to handling off-policy learning that combines a form of model learning with a new objective function. Minimizing the objective function is show to result in lower variance estimates versus other typical approaches such as importance sampling. Overall the paper was pretty good, but a little hard to follow as it feels written more in the style of papers trying to estimate treatment effects than in the style of typical RL papers. Further along these lines, there was not distinctly specified algorithm in the paper. This does not imply that one could not be inferred from the math, but it is something of an obstacle for readers who approach problems more algorithmically. Some assumptions seem a bit strong, e.g., that phi is twice differentiable. It would be nice to see some more discussion of that. The presentation is a bit confusing because the paper references things in the supplemental material as if they were part of the main paper. For example, there is a reference to lemma, but this doesn’t appear in the main body of the paper. It would be helpful to make the main paper more self contained with explicit and clear references to things in supplemental materials where approach. As it is, it reads as a paper that has been roughly ripped into the two pieces without any clean up afterwards. The empirical evaluation involved two domains, one that is very familiar and easy - inverted pendulum - and another that is less familiar and probably quite difficult. It might have been nice to see an intermediate problem that would help give more insight into what is going on. The experimental results which are presented are quite good and encouraging. Overall, think this is a decent paper with a few rough spots. Minor: “for an specific” -> “for a specific”